health and disease and epidemiology/theoretical biology/biomathematics

disease ecology, complex networks, multi-pathogen systems, epidemic spreading

**Author for correspondence:**
Chiara Poletto
e-mail: chiara.poletto@inserm.fr

# Interplay between competitive and cooperative interactions in a three-player pathogen system

Francesco Pinotti[1], Fakhteh Ghanbarnejad[2,3,4], Philipp Hövel[2,5] and Chiara Poletto[1]

[1]INSERM, Sorbonne Université, Institut Pierre Louis d'Épidémiologie et de Santé Publique, IPLESP, Paris 75012, France
[2]Institut für Theoretische Physik, Technische Universität Berlin, Hardenbergstraße 36, Berlin 10623, Germany
[3]The Abdus Salam International Centre for Theoretical Physics (ICTP), Trieste, Italy
[4]Physics Department, Sharif University of Technology, PO Box 11165-9161, Tehran, Iran
[5]School of Mathematical Sciences, University College Cork, Western Road, Cork T12 XF62, Republic of Ireland

CP, 0000-0002-4051-1716

In ecological systems, heterogeneous interactions between pathogens take place simultaneously. This occurs, for instance, when two pathogens cooperate, while at the same time, multiple strains of these pathogens co-circulate and compete. Notable examples include the cooperation of human immunodeficiency virus with antibiotic-resistant and susceptible strains of tuberculosis or some respiratory infections with *Streptococcus pneumoniae* strains. Models focusing on competition or cooperation separately fail to describe how these concurrent interactions shape the epidemiology of such diseases. We studied this problem considering two cooperating pathogens, where one pathogen is further structured in two strains. The spreading follows a susceptible-infected-susceptible process and the strains differ in transmissibility and extent of cooperation with the other pathogen. We combined a mean-field stability analysis with stochastic simulations on networks considering both well-mixed and structured populations. We observed the emergence of a complex phase diagram, where the conditions for the less transmissible, but more cooperative strain to dominate are non-trivial, e.g. non-monotonic boundaries and bistability. Coupled with community structure, the presence of the cooperative pathogen enables the coexistence between strains by breaking the spatial symmetry and dynamically creating different ecological niches. These results shed light on ecological mechanisms that may impact the epidemiology of diseases of public health concern.

# 1. Introduction

Pathogens do not spread independently. Instead, they are embedded in a larger ecosystem that is characterized by a complex web of interactions among constituent elements. Among ecological forces shaping such ecosystems, pathogen–pathogen interactions have drawn increasing attention during recent years owing to their population-level impact and public health consequences. Recent advances in serological tests and genotyping techniques have improved our reconstruction of pathogen populations where multiple strains co-circulate, often competing owing to cross-protection or mutual exclusion. Examples include tuberculosis [1,2], *Plasmodium falciparum* [3], *Streptococcus pneumoniae* [4,5] and *Staphylococcus aureus* [6,7]. Polymorphic strains can also interact in more complex ways, with both competition and cooperation acting simultaneously, as observed in co-circulating dengue serotypes [8]. While interfering with each other, strains also interact with other pathogens co-circulating in the same population. Tuberculosis [1], human papillomavirus (HPV) [9] and *P. falciparum* [10], for example, appear to be facilitated by human immunodeficiency virus (HIV), whereas *Str. pneumoniae* benefits from some bacterial infections, e.g. *Moraxella catarrhalis*, and is negatively associated with others such as *Sta. aureus* [11,12]. Competition, cooperation and their co-occurrence may fundamentally alter pathogen persistence and diversity, thus calling for a deep understanding of these forces and their quantitative effects on spreading processes.

Mathematical models represent a powerful tool to assess the validity and impact of mechanistic hypotheses about interactions between pathogens or pathogenic strains [13,14]. The literature on competitive interactions is centred on pathogen dominance and coexistence. Several factors were found to affect the ecological outcome of the competition, including co-infection mechanisms [15–18], host age structure [19,20], contact network [21–32] and spatial organization [33–38]. At the same time, models investigating cooperative interactions have driven many research efforts during recent years [39–45]. Cooperation has been found to trigger abrupt transitions between disease extinction and large scale outbreaks along with hysteresis phenomena where the eradication threshold is lower than the epidemic threshold [39,40,43]. These findings were related to the high burden of synergistic infections, e.g. the HIV and tuberculosis co-circulation in many parts of the world. Despite considerable mathematical and computationally heavy research on interacting pathogens, competition and cooperation have been studied mostly separately. Nevertheless, current understandings about these mechanisms taken in isolation may fail to describe the dynamics arising from their joint interplay, where heterogeneous interactions may shape the phase diagram of coexistence/dominance outcome, along with the epidemic prevalence.

Here we studied the simplest possible epidemic situation where these heterogeneous effects are at play. We introduced a three-player model where two pathogens cooperate, and one of the two is structured in two mutually exclusive strains. This mimics a common situation, where e.g. resistant and susceptible strains of *Str. pneumoniae* cooperate with other respiratory infections [11] and allows us to address two important ecological questions:

— how does the interplay between two distinct epidemiological traits, i.e. the transmissibility and the ability to exploit the synergistic pathogen, affect the spreading dynamics? and
— how does the presence of a synergistic infection alter the coexistence between competing strains?

We addressed these questions by providing a characterization of the phase space of dynamical regimes. We tested different modelling frameworks (continuous and deterministic vs. discrete and stochastic) and compared two assumptions regarding population mixing, i.e. homogeneous vs. community structure.

The paper is structured as follows: §2 introduces the main aspects of the three-player model. We provide the results of the deterministic dynamical equations in §3.1, where we present the stability analysis, together with the numerical integration of the equations, to characterize the phase space of the dynamics. The structuring of the population in two communities is analysed in §3.2. In §3.3, we describe the results obtained within a network framework comparing stochastic simulations in an Erdős–Rényi and a random modular network. We discuss the implications of our results in §4.

# 2. The model

A scheme of the model is depicted in figure 1*a*. We considered the case in which two pathogens, *A* and *B*, follow susceptible-infected-susceptible (SIS) dynamics, and we made the simplification that they both have the same recovery rate $\mu$. *A* and *B* cooperate in a symmetric way through increased

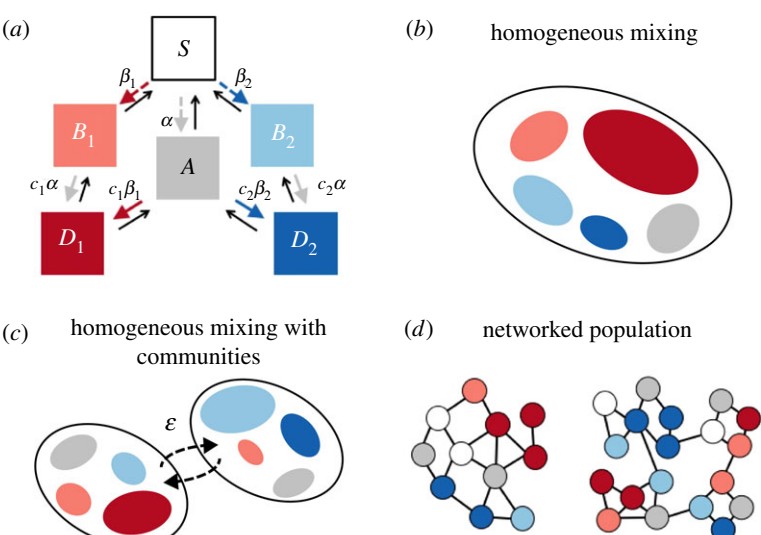

**Figure 1.** Scheme of the model. (*a*) Compartmental model. Coloured arrows represent transitions occurring owing to infection transmission. Dashed arrows refer to primary infections, while solid arrows refer to secondary ones; transmission parameters are also reported close to each arrow. Black arrows represent recovery transitions. (*b–d*) Schematic representation of the modelling frameworks and population structures considered. (*b*) A homogeneously mixed population (§3.1). (*c*) Two homogeneous populations with across-group mixing ruled by the parameter $\varepsilon$ (§3.2); in (*b*),(*c*), colours indicate the infectious density for each compartment. (*d*) Erdős–Rényi and random modular networks (§3.3). Colours indicate the nodes' status.

susceptibility, i.e. a primary infection by one of the two increases the susceptibility to a secondary infection by the other pathogen. We assumed that the cooperative interaction does not affect infectivity, thus doubly infected individuals, i.e. infected with both $A$ and $B$, transmit both diseases at their respective infection rates. $B$ is structured in two strains, $B_1$ and $B_2$, that compete through mutual exclusion (co-infection with $B_1$ and $B_2$ is impossible) and differ in epidemiological traits. Specifically, we denoted the infection rates for pathogens $A$ and $B_i$ with $\alpha$ and $\beta_i$ ($i = 1, 2$), respectively. We introduced the parameters $c_i > 1$ to represent the increased susceptibility after a primary infection. In summary, individuals can be in either one of six states: susceptible ($S$), singly infected ($A$, $B_i$) and doubly infected with both $A$ and $B_i$. The latter status is denoted by $D_i$.

To simplify the analytical expressions, we rescaled time by the average infectious period $\mu^{-1}$, which leads to non-dimensional equations. The basic reproductive ratios of each player, $R_0^{(i)} = \beta_i/\mu$ and $R_0^{(A)} = \alpha/\mu$, then become equal to the transmission rates $\beta_i$ and $\alpha$, respectively. This implies that the threshold condition $\beta_i$, $\alpha > 1$ has to be satisfied in order for the respective player to be able to individually reach an endemic state. Assuming a homogeneously mixed population, the mean-field equations describing the spreading dynamics are:

$$
\left.
\begin{aligned}
\dot{S} &= A + B_1 + B_2 - \alpha S X_A - \beta_1 S X_1 - \beta_2 S X_2 \\
\dot{B}_1 &= D_1 - B_1 - c_1 \alpha B_1 X_A + \beta_1 S X_1 \\
\dot{B}_2 &= D_2 - B_2 - c_2 \alpha B_2 X_A + \beta_2 S X_2 \\
\dot{A} &= D_1 + D_2 - A + \alpha S X_A - c_1 \beta_1 A X_1 - c_2 \beta_2 A X_2 \\
\dot{D}_1 &= -2 D_1 + c_1 \alpha B_1 X_A + c_1 \beta_1 A X_1 \\
\dot{D}_2 &= -2 D_2 + c_2 \alpha B_2 X_A + c_2 \beta_2 A X_2,
\end{aligned}
\right\}
\tag{2.1}
$$

and

where the dot indicates a differentiation with respect to time rescaled by $\mu^{-1}$, and quantities $S$, $A$, $B_i$ and $D_i$ represent occupation numbers of the compartments divided by the population. The variables $X_A$, $X_i$, $i = 1, 2$, indicate the total fractions of individuals carrying $A$ and $B_i$, respectively, among the singly and doubly infected individuals. They satisfy the equations:

$$
\dot{X}_i = X_i \beta_i (S + c_i A) - X_i
\tag{2.2a}
$$

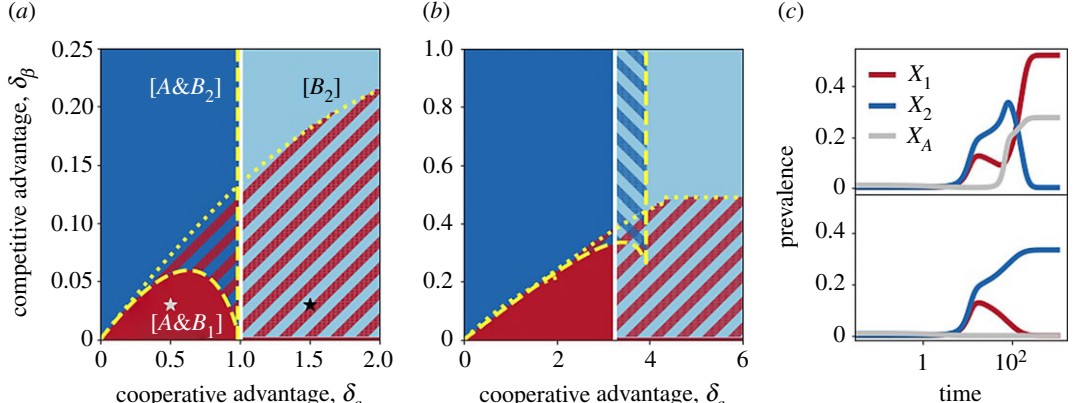

**Figure 2.** Phase diagram for the well-mixed system. (*a,b*) Stable equilibria as a function of $\delta_\beta$ and $\delta_c$ for two parameter choices, namely (*a*) $\alpha = 0.6$, $\beta_2 = 1.5$, $c_1 = 4$ and (*b*) $\alpha = 0.8$, $\beta_2 = 1.1$, $c_1 = 7$. The three states $[B_2]$, $[A\&B_2]$ and $[A\&B_1]$ are indicated in light blue, dark blue and red, respectively. Hatched regions correspond to bistable and multi-stable regions. The yellow curves show the analytical boundaries delimiting stability regions for $[A\&B_1]$ and $[A\&B_2]$ (equations (3.2) and (3.3)), while the white one delimits the $[B_2]$'s region (equation (3.1)). Note that for $\delta_c > 3, 6$, for (*a,b*), respectively, $c_2 < 1$ and the interaction between $B_2$ and $A$ ceases to be cooperative. This naturally provides a range for the *x*-axis. In (*b*), $\beta_1$ is below one for $\delta_\beta > 0.1$. (*c*) Evolution of total prevalence for $A$ (grey), $B_1$ (red) and $B_2$ (blue), considering singly and doubly infected combined. Parameters correspond to the grey and black star markers in (*a*), i.e. $\delta_\beta = 0.03$ and $\delta_c = 0.5, 1.5$ in top and bottom panels, respectively. Dynamical trajectories have been obtained by integrating equations (2.1) with initial conditions: $B_i(t = 0) = 0.001$, $A(t = 0) = 0.01$.

and

$$\dot{X}_A = X_A \alpha(S + c_1 B_1 + c_2 B_2) - X_A. \tag{2.2b}$$

Without the loss of generality, we considered the case in which the strain $B_2$ is more transmissible than $B_1$, i.e. $\delta_\beta = \beta_2 - \beta_1 > 0$. Furthermore, we focused on the more interesting case of trade-off between transmissibility and cooperation to limit the parameter exploration: the less transmissible strain, $B_1$, is more cooperative, $\delta_c = c_1 - c_2 > 0$. If $B_2$ is more cooperative, we expect it to win the competition. To summarize, our main assumptions are:

— $\delta_\beta = \beta_2 - \beta_1 > 0$,
— $\delta_c = c_1 - c_2 > 0$,
— $c_i > 1$ $i = 1, 2$.

In §3, we will first describe the dynamics arising from the deterministic equation (2.1). We will then consider the case in which the whole population is structured in two groups (figure 1*c*). Finally, we will apply the proposed model to contact networks, where nodes represent individuals and transmission occurs through links, and consider transmission and recovery as stochastic processes. Two types of networks will be tested: Erdős–Rényi and random modular networks (figure 1*d*).

# 3. Results

## 3.1. Continuous well-mixed system

We carried out a stability analysis to classify the outcome of the interaction as a function of the difference in strain epidemiological traits, $\delta_c$ and $\delta_\beta$. Specifically, we computed explicit analytical expressions for states' feasibility and stability conditions in several cases. Furthermore, we performed extensive numerical simulations in cases where closed expressions were difficult to obtain. We present the overall behaviour and the main analytical results in this section and we refer to the electronic supplementary material for the detailed calculations. In the following, we will use square brackets to indicate final state configurations in terms of persisting strains, thus $[A\&B_1]$ indicates, for instance, the equilibrium configuration where both $A$ and $B_1$ persist, while $B_2$ becomes extinct.

Figure 2*a,b* shows the location of stable states with two combinations of $\alpha$, $\beta_2$ and $c_1$. Results that are obtained for other parameter values are reported in the electronic supplementary material, figure S1. No coexistence was found between $B_1$ and $B_2$. In principle, equations (2.1) admit a coexistence equilibrium

[$A\&B_1\&B_2$]. However, this coexistence was always found to be unstable in the numerical simulations. Persistence of $A$ is only possible together with one of the $B$ strains. The equilibrium solution [$A$] is unfeasible for $\alpha < 1$ and unstable for $\alpha > 1$, unless both reproductive ratios, $\beta_i$, are below the epidemic threshold. Because of the assumption $\delta_\beta > 0$, $B_2$ outcompetes $B_1$ in absence of $A$, in agreement with the principle of competitive exclusion. Therefore, the final state [$B_1$] is always unstable, and persistence of $B_1$ is possible only in co-circulation with $A$. On the other hand, $B_2$ can spread either alone or together with $A$. Specifically, the [$B_2$] configuration is feasible for $\beta_2 > 1$. It is stable if and only if $\alpha < \alpha_c$, with

$$\alpha_c = \frac{\beta_2}{c_2(\beta_2 - 1) + 1}. \tag{3.1}$$

This provides a sufficient condition for the persistence of $A$. Equation (3.1) can be expressed in terms of $\delta_c$, namely $\delta_c > c_1 - (\beta_2 - \alpha)/[\alpha(\beta_2 - 1)]$, which is visualized as the white boundary in figure 2a,b.

The competition between $B_1$ and $B_2$ is governed by the trade-off between transmission and cooperative advantage. This is described by the boundaries of the [$A\&B_i$] regions that can be traced by combining the feasibility and stability conditions. These boundaries are plotted in figure 2a,b as dotted and dashed yellow curves for [$A\&B_1$] and [$A\&B_2$], respectively. For a solution to be feasible, the densities of all states must be non-negative. For absolute parameter values as in figure 2a,b we found that this yields the necessary condition $\alpha\beta_i > 4(c_i - 1)/c_i^2$, corresponding to the vertical and horizontal segments. On the other hand, the stability boundary separating [$A\&B_i$] from any state containing $B_j$ ($j \neq i$) is given by

$$\beta_j(S^* + c_j A^*) - 1 < 0, \tag{3.2}$$

where $S^*$ and $A^*$ are the equilibrium densities of $S$ and $A$, respectively, evaluated in the configuration [$A\&B_i$]. The left-hand side of the equation represents the growth rate of the competitor $B_j$, appearing in equation (2.2a), and evaluated in the [$A\&B_i$] state. Thus, the relation (3.2) expresses the condition for $B_j$ extinction. Expressed in terms of $\delta_c$ and $\delta_\beta$, the conditions become:

$$\left.\begin{aligned}
[A\&B_1] &: \frac{\beta_2(c_1 - \delta_c)}{c_1(\beta_2 - \delta_\beta)} + \frac{\beta_2 \delta_c}{c_1 - 1}\left(1 - \sqrt{1 - \frac{4(c_1 - 1)}{c_1^2(\beta_2 - \delta_\beta)\alpha}}\right) = 1 \\
\text{and} \qquad [A\&B_2] &: \frac{c_1(\beta_2 - \delta_\beta)}{\beta_2(c_1 - \delta_c)} - \frac{\delta_c(\beta_2 - \delta_\beta)}{c_1 - \delta_c - 1}\left(1 - \sqrt{1 - \frac{4(c_1 - \delta_c - 1)}{\beta_2 \alpha(c_1 - \delta_c)^2}}\right) = 1.
\end{aligned}\right\} \tag{3.3}$$

The intersection among the stability boundaries described above produces a rich state space. For all tested values of $\alpha$, $\beta_2$ and $c_1$, we found a wide region of the $(\delta_c, \delta_\beta)$ space (red-hatched in figure 2a,b) displaying bistability between the [$A\&B_1$] state and a $B_2$-dominant state with either [$B_2$] or [$A\&B_2$]. In certain cases, bistability can also occur between the [$B_2$] and [$A\&B_2$] states (blue-hatched region in figure 2b). This has been studied in the past for two cooperating pathogens [43]. We found that the intersection between the latter region and the red-hatched region gives rise to a multi-stable state.

Interestingly, for all tested parameters, we found that the boundary of the [$A\&B_2$] stability region is not monotonic. As a consequence, for a fixed $\delta_\beta$, a first transition from the [$A\&B_2$] state to [$A\&B_1$] is found for small $\delta_c$ values. An increase of $\delta_c$ leads to a second boundary with a bistable region, where the dominance of $B_1$ over $B_2$ depends on initial conditions. The transition for small $\delta_c$ is expected: by increasing $B_1$'s advantage in cooperation, a point is reached beyond which $B_1$'s disadvantage in transmissibility is overcome. On the other hand, the second threshold appears to be counterintuitive. We investigated it more in depth for the case depicted in figure 2a. We plotted the infectious population curves as a function of time for each infectious compartment. We compared $\delta_c = 0.5$, which corresponds to the [$A\&B_1$] stable state (figure 2c top), and $\delta_c = 1.5$, which leads to a bistable region (figure 2c bottom), where all other parameters are as in figure 2a. Figure 2c shows that $B_1$ loses the competition at the beginning. However, when $B_2$ is sufficiently cooperative with $A$ (top), the rise of $B_2$ leads to a rise in $A$ that ultimately drives $B_1$ to dominate. For higher $\delta_c$, the strength of cooperation between $B_2$ and $A$ is not sufficient. The indirect beneficial effect of $B_2$ over $B_1$ is not present (bottom), and $B_1$ can dominate only if initial conditions are favourable.

In the bistable and multi-stable regions, the outcome of the competition is determined by initial conditions. While a mathematical analysis is complicated owing to the multi-dimensionality of the problem, we gained insights into the basins of attraction by numerically integrating equation (2.1) while exploring different combinations of $B_i(t = 0)$ and $A(t = 0)$. For the bistability between the regions

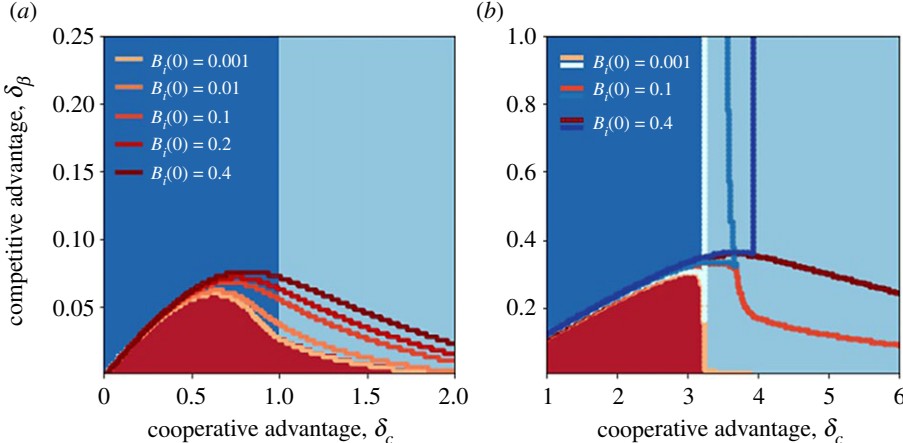

**Figure 3.** Equilibrium configurations for the well-mixed system. Final outcome obtained by numerically integrating equations (2.1) for $B_i(t=0) = 0.001$, $A(t=0) = 0.01$. (a) $\alpha = 0.6$, $\beta_2 = 1.5$, $c_1 = 4$. Boundaries of the $[A\&B_1]$ state for different initial conditions are indicated by red-scale contours. (b) $\alpha = 0.8$, $\beta_2 = 1.1$, $c_1 = 7$. Here, the boundaries of the $[A\&B_2]$ are shown (in blue shades), together with the ones of $[A\&B_1]$.

of $B_1$ and $B_2$ dominance, we considered the parameter combination of figure 2$a$ and shown in figure 3$a$ the states that are reached starting from $B_i(t=0) = 0.001$ and $A(t=0) = 0.01$. The bundle of curves with different shades of red (from light to dark) indicates the boundary of the $[A\&B_1]$ equilibrium when $B_1(t=0)$ and $B_2(t=0)$ are equally increased. We found that an increase in $B_1$'s initial infected densities favours the $[A\&B_1]$ state, as expected. Interestingly, however, an increase in $B_1(t=0)$ results in the $[A\&B_1]$ region expanding even when $B_2$'s density increases at the same level. Figure 3$b$ shows that a similar behaviour is found when parameters are as in figure 2$b$. In this case, the region $[A\&B_2]$ expands together with the $[A\&B_1]$ one. Thus, increased initial frequencies promote co-circulation between $B$ and $A$. In the electronic supplementary material, figure S2, we present a deeper exploration of initial conditions, considering the parameter combination of figure 2$a$ as an example. We found that an increase in the initial level of $A$ also favours $B_1$. However, the initial advantage (either in $B_1(0)$ or $A(0)$) that is necessary for $B_1$ to win against $B_2$ increases as $\delta_\beta$ increases.

The stability diagrams obtained with several parameter sets, explored in a latin-square fashion, is reported in the electronic supplementary material, figure S1. This shows that increased transmissibility and cooperativity levels enhance the cooperative interaction of $B_i$ strains with $A$. This results in an increase in the parameter region for which $B_1$ together with $A$ dominates over $B_2$. For instance, the comparison between figure 2$d$ and $f$ shows that, by increasing $\beta_2$ from 1.1 to 1.5, the same difference in strain epidemiological traits, $\delta_c$ and $\delta_\beta$, may lead to a switch in dominance from $B_2$ to $B_1$.

## 3.2. Continuous system with communities

We now consider a population that is divided into two communities (cf. figure 1$c$). For simplicity, we assumed that they are of the same size. To differentiate transmission within and across communities, we rescaled the force of infection produced by individuals of a different community by a factor $\varepsilon$, and the force of infection of individuals of the same community by $1 - \varepsilon$. We assumed $0 < \varepsilon \le \frac{1}{2}$ in order to consider the case in which individuals mix more within their community than outside – the limit $\varepsilon = \frac{1}{2}$ corresponds to homogeneous mixing.

Given the high number of variables, a stability analysis is difficult in this case. Still, the dynamics can be reconstructed through numerical integration of the equations. Figure 4 shows the final states with fixed $\varepsilon$, $\beta_2$, $C_1$ and $\alpha$. Other parameter values are analysed in the electronic supplementary material, figure S3. Figure 4$a$–$c$ compares different seeding configurations, while keeping the initial density of each pathogen/strain to 0.01: (a) all strains are seeded in community 1 and community 2 is completely susceptible; (b) $B_1$ is seeded in community 1 while $B_2$ and $A$ are initially present in community 2 only; and (c) $B_1$ and $A$ are seeded together in community 1, while $B_2$ is seeded in community 2. In all cases, we found a diagram with a shape similar to figure 3$a$. However, a new region is now present (indicated in black) where all players coexist. This occurs when strains are separated since the beginning—see the electronic supplementary material, figure S4 for additional

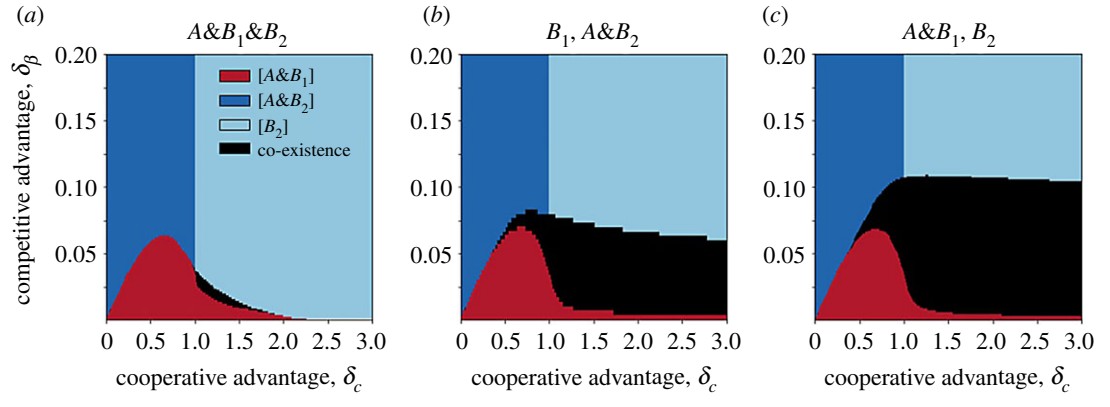

**Figure 4.** Equilibrium configurations for two interacting communities. Final outcome obtained by numerically integrating the equations when: (*a*) all strains start in the same community (together with *A*); (*b*) $B_1$ and $B_2$ start in separate communities, with *A* starting together with $B_2$; (*c*) *A* starts along with $B_1$, while $B_2$ starts separately. Initial density of each pathogen/strain is 0.01. Here $\varepsilon = 0.0002$. Other parameters are as in figure 2*a*.

seeding configurations. Interestingly, however, this also happens for a tiny region of the parameter space, when all strains are seeded together (figure 3*a*), provided that the other community is initially disease-free.

Figure 5 sheds light on the dynamics leading to the outcomes of figure 4. In order to benefit from the cooperative advantage, the $B_1$ incidence must be above a certain threshold. Figure 5*b*,*c* show that the incidence of *A* remains close to zero, until the incidence of $B_1$ is sufficiently high. With $B_2$ seeded on a different community (community 2), the direct interaction between the two strains is delayed by the time necessary for $B_2$ to reach the community of $B_1$. For high $\varepsilon$, the delay is short and $B_2$ reaches community 1 before *A* incidence starts to rise (figure 5*a*). On the other hand, for lower $\varepsilon$, $B_1$ has enough time to build up a cooperative protection before the arrival of $B_2$. This makes it resistant to the invader. At intermediate $\varepsilon$, $B_1$ becomes able to overcome $B_2$ in community 2. For small $\varepsilon$, strains spread in their origin community independently from one another.

In summary, a decrease in $\varepsilon$ increases the region of $B_1$ persistence (figure 5*d*). However, this may be associated with either $B_1$ dominance or coexistence. Reducing the values of $\varepsilon$, the region corresponding to the [$A\&B_1$] state expands first and shrinks later, leaving the place to the coexistence region. This is shown by the non-monotonous change of the [$A\&B_1$] region in figure 5*e*.

When all strains start in the same community, coexistence is enabled by a segregation mechanism similar to the one described above. In this case, separation occurs during the early stage: $B_2$ rapidly spreads in the other community owing to its advantage in transmissibility and becomes dominant there (cf. electronic supplementary material, figure S5). This enables coexistence in a parameter region where $B_1$ would otherwise dominate.

Results described so far were obtained with fixed values of $\beta_2$, $C_1$ and $\alpha$. Additional parameter choices are shown in the electronic supplementary material, figure S3. Increasing in $\alpha$ was found to enlarge the $B_1$ dominance region, as in the well-mixed case. In addition, coexistence becomes possible for $\alpha > 1$ in a very small region of the parameter space.

## 3.3. Spreading on networks

The continuous-deterministic framework analysed so far does not account for stochasticity and for the discrete nature of individuals and their interactions. These aspects may alter the phase diagram and shape the transitions across various regions. We cast our model on a discrete framework in which individuals are represented by nodes in a static network. Possible individual states are still the same as in the mean-field formulation, and infection can spread only between neighbouring nodes. We first considered an Erdős–Rényi graph, where the mixing is homogeneous across nodes. Denoting $N$ the number of nodes and $\bar{k}$ the average degree, the network was built by connecting any two nodes with probability $\bar{k}/(N-1)$. We run stochastic simulations of the dynamics. In order to see the effect of multi-pathogen interactions, we minimized the chance of initial stochastic extinction by infecting a relatively high number of nodes at the beginning: 100 infected for each infectious agent. We then computed the fraction of stochastic simulations ending up in any final state, the average prevalence

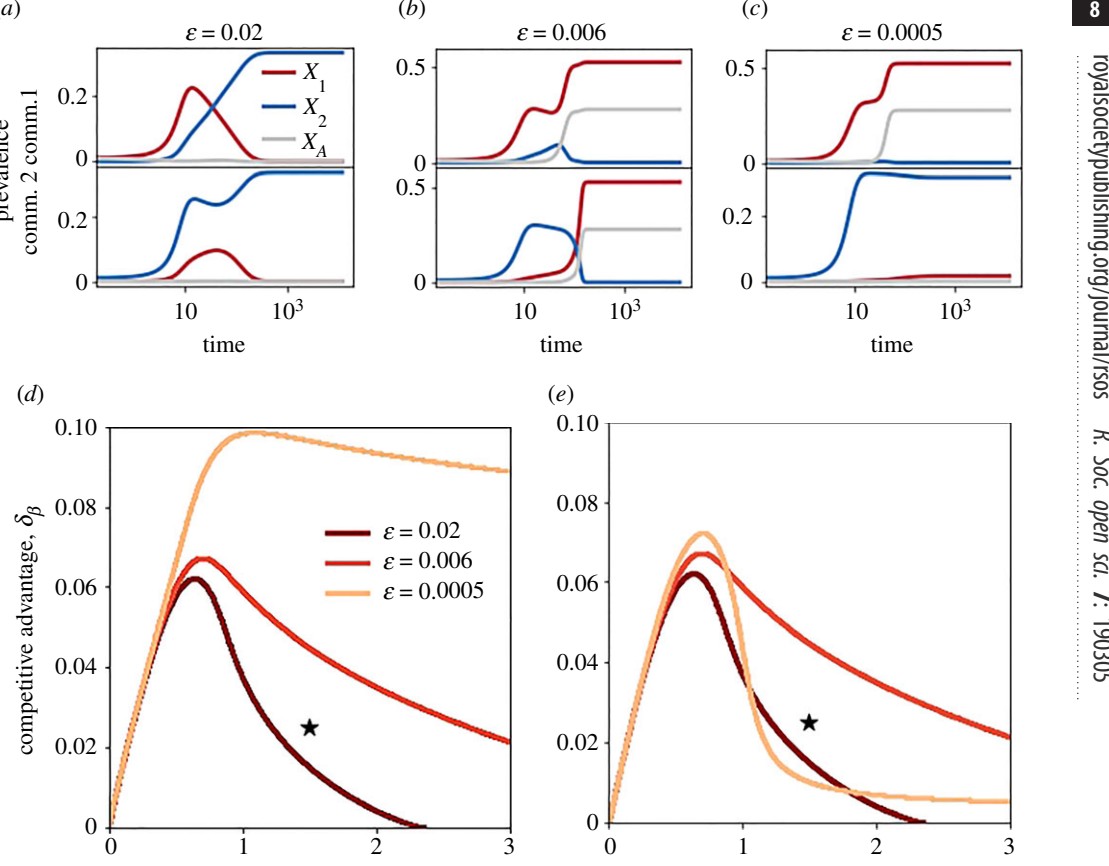

**Figure 5.** Role of spatial separation for two interacting communities. (*a*–*c*) Dynamical trajectories within communities 1 and 2 obtained for (*a*) $\varepsilon = 0.02$, (*b*) $\varepsilon = 0.006$ and (*c*) $\varepsilon = 0.0005$. (*d,e*) Boundaries in the $\delta_c$, $\delta_\beta$ plane delimiting the regions where the dynamics ends up in (*d*) $B_1$ persistence (i.e. [$A\&B_1$] or full coexistence); (*e*) [$A\&B_1$] state. In all panels, $A$ and $B_1$ are seeded together into one community, while $B_2$ is seeded into the other community; the initial density of each species is set to 0.01. Trajectories are obtained by setting $\delta_c = 1.5$, $\delta_\beta = 0.025$ (black star in (*d,e*)). Other parameters are as in figure 2*a*.

for each strain ($X_1$, $X_2$) in the final state and the average coexistence time. Additional details on the network model and the simulations are reported in the electronic supplementary material.

The phase diagram of figure 6*a* is similar in many aspects to its continuous deterministic version (figure 3*a*). Three final states are possible, i.e. [$B_2$], [$A\&B_1$] and [$A\&B_2$] (figure 6*a*). Here, however, the same initial conditions and parameter values can lead to different stochastic trajectories and stationary states. For instance, the red region in the figure corresponds to the case in which the final state [$A\&B_1$] is reached very frequently; however, the dynamic trajectories can also end up in the [$A\&B_2$] or in the [$B_2$] states. The transitions across the different regions of the diagrams can be very different, as demonstrated by figure 6*b*–*j*. Figure 6*b*–*d* shows the effect of varying $\delta_c$ at a fixed $\delta_\beta$. The transition between [$A\&B_2$] and [$A\&B_1$] on the left is sharp. Both the probability of one strain winning over the other and the equilibrium prevalence change abruptly for a critical value of $\delta_c$. Here, the spreading is super-critical for all pathogens: $\beta_1$, $\beta_2 > 1$ and $c_1$, $c_2$ are sufficiently high to sustain the spread of $A$. The transition is owing to the trade-off between $B_1$ and $B_2$ growth rates. Conversely, the probability of ending up in the [$B_2$] state rises slowly, driving the gradual transition from the red to the light blue region on the right. This region appears in correspondence of the bistable region of the continuous/ deterministic diagram: figure 2. Here, $A$ undergoes a transition from persistence to extinction, driven by the drop in $c_2$ (electronic supplementary material, figure S6). This critical regime is characterized by enhanced stochastic fluctuations. When $\delta_c$ is fixed and $\delta_\beta$ varies, we found a sharp transition (figure 6*e*–*g*) and a hybrid transition, where the final state probability varies gradually and the equilibrium prevalence ($X_1$) varies abruptly (figure 6*h*–*j*).

We concluded by analysing the effect of community structure. Each node was assigned to one among $n_C$ communities, which we assumed for simplicity to have equal size $N/n_C$, and has a number of open

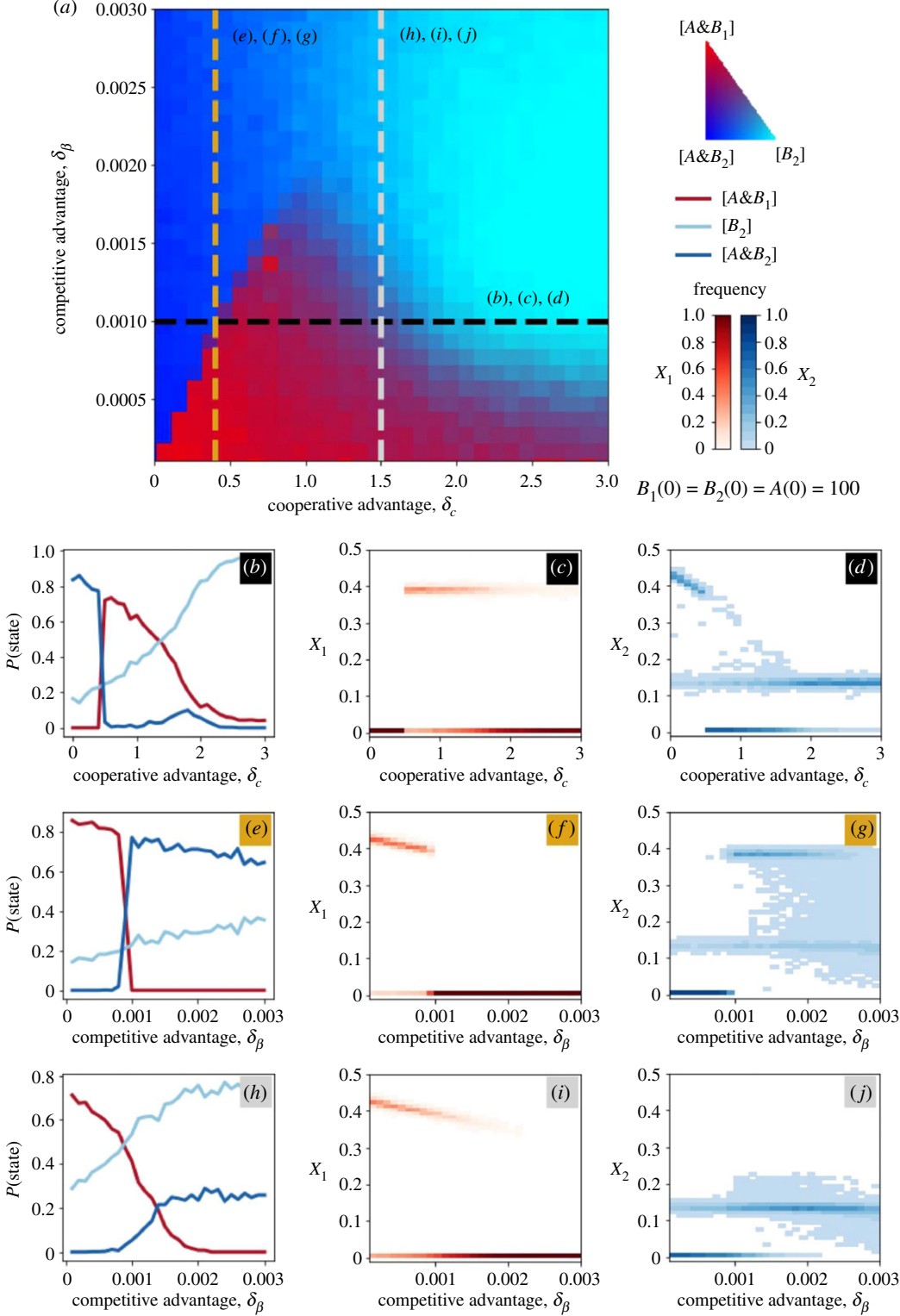

**Figure 6.** Phase diagram for the Erdős–Rényi network. (*a*) Frequency of stationary states, as obtained in numerical simulations. The colour scale in the legend quantifies the proportion of runs ending in the different states among $[B_2]$, $[A\&B_2]$ and $[A\&B_1]$. Here, the extremes of the colour map correspond to the case in which these states are found in 100% of runs. Initial conditions are shown in the figure. (*b–j*) Equilibrium state probability (left column), and distribution of both $B_1$'s and $B_2$'s prevalence in the final state (middle and right columns respectively) along the dashed lines in (*a*). Specifically: $\delta_\beta = 0.001$ for (*b–d*); $\delta_c = 0.4$ for (*e–g*); $\delta_c = 1.5$ for (*h–j*). For convenience, we reparametrized the model taking the time step as unit of time: $\Delta t = 1$. We set the following parameters' values: $\mu = 0.05$, $\alpha = 0.009$, $\beta_2 = 0.015$, $c_1 = 4$, $N = 20\,000$, $\bar{k} = 4$. Note that in the case of spread on networks, we have $R_0^{(i)} = \beta_i \rho / \mu$ and $R_0^{(A)} = \alpha \rho / \mu$, where $\rho$ is the spectral radius of the adjacency matrix. For the $\alpha$, $\beta_2$ and $\delta_\beta$ values considered in the figure, we have $R_0^{(1)}$, $R_0^{(2)} > 1$.

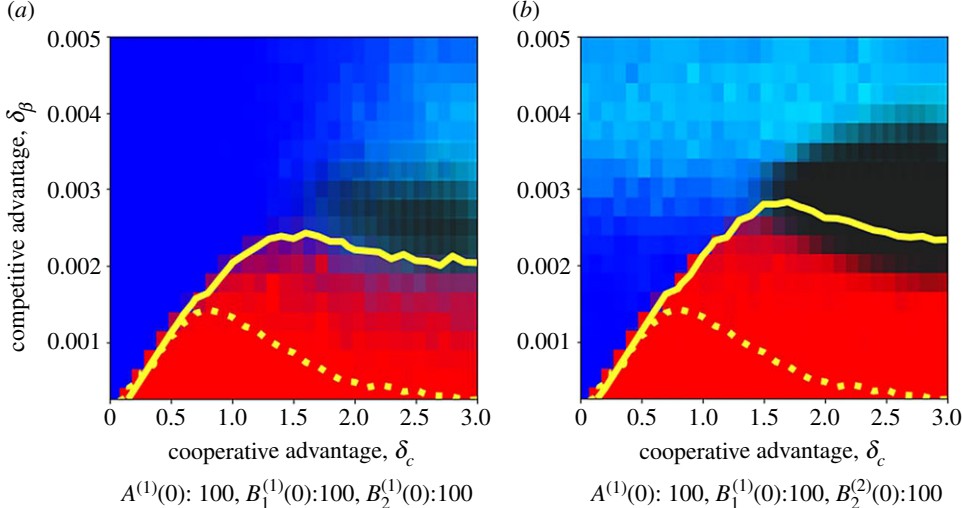

**Figure 7.** Phase diagram for the random modular network. Frequency of equilibrium configurations, as obtained in the numerical simulations, with (a) $A$, $B_1$ and $B_2$ starting in the same community, and (b) $A$ and $B_1$ starting from a different community to $B_2$. Detailed initial conditions are directly shown on each panel. For each player, the superscript $i$ indicates the community where the infectious are seeded. The colour scale is the same as in figure 6. The frequency of runs for which coexistence of all strains was observed after $T_{max} = 2 \times 10^6$ time steps is shown with different shades of black. Contour lines representing the 0.5 probability to end up in the $[A \& B_1]$ state are indicated to enable a comparison between the Erdős–Rényi (dashed line) and the random modular network (continuous line). We considered $n_C = 10$ and $\varepsilon = 0.003$. Other parameter values are as in figure 6.

connections drawn from a Poisson distribution with average $\bar{k}$. Links were formed by matching these connections according to an extended configuration model, where a fraction $\varepsilon$ of stubs connects nodes of different communities. In this way, the model is the discrete version of the one in §3.2.

Mean-field results remain overall valid. The two plots in figure 7 mirror figure 4a,c and show a similar behaviour. We find evidence of a coexistence region (in black in the figure), where no extinction is observed during the simulation time frame—here set to $2 \times 10^6$ time steps, around two orders of magnitude longer than the time needed to observe strain extinction in the Erdős–Rényi case. Such a region is larger when the two strains are seeded in separated communities (figure 7b), but it is still visible when strains start all together (figure 7a). Coexistence occurs less frequently in the latter case because it requires strains to reach the separation during the spreading dynamics.

Analogously to the continuous deterministic model, we found that the separation in communities favours the more cooperative strain. The region where $B_1$ wins is larger compared to the Erdős–Rényi case (as highlighted by the comparison between the dashed and the continuous curves). In addition, the probability of winning is close to one for a large portion of the $[A \& B_1]$ dominance region.

# 4. Discussion and conclusion

We presented here a theoretical analysis of a three-player system where both competition and cooperation act simultaneously. We have considered two competing strains co-circulating in the presence of another pathogen cooperating with both of them. Strains differ in epidemiological traits, with one strain being more transmissible but less cooperative than its competitor. Through mathematical analyses and computer simulations, we have reconstructed the possible dynamical regimes, quantifying the conditions for dominance of one strain or coexistence. We found that the interplay between competition and cooperation leads to a complex phase diagram whose properties cannot be easily anticipated from previous works that considered competition and cooperation separately.

We showed that it is possible for a more cooperative strain to dominate over a more transmissible one, provided that the difference in transmissibility is not too high. This suggests that the presence of another pathogen ($A$) might alter the spreading conditions, creating a favourable environment for a strain that would be otherwise less fit. While dominance depends on the difference in epidemiological traits, we found that variations in the absolute cooperation and transmissibility levels may change the hierarchy

between strains—analogously to [15]—with a higher spreading potential of either $B_i$ or $A$ favouring the more cooperative strain.

Interestingly, the cooperative strain can also dominate when $A$ has a sub-critical reproductive ratio ($\alpha < 1$)—when spreading alone—and relies on the synergistic interaction with $B$ strains to persist. The dynamical mechanisms underlying this outcome are complex. We analysed a case with a small difference in cooperativity, and we found that the more transmissible strain, by spreading initially faster, creates the bulk of $A$ infections that in turn favour its competitor. In other words, direct competition for susceptible hosts is not the only force acting between strains: an indirect, beneficial interaction is also at play, mediated by the other pathogen. The dominance outcome is thus the result of the trade-off between these two forces. When the difference in cooperation is higher, two or more stationary configurations are possible. In this scenario, the final outcome is also determined by the initial frequency of each pathogen/strain. We found that, in certain situations, an initial advantage of one strain is able to drive it to dominance. This is in contrast with simpler models of competition, where the final outcome is determined solely by the epidemiological traits. The outcome, however, is also governed by pathogen $A$ that favours the more cooperative strain. Previous works have analysed multistability in two-pathogen models with cooperation in relation to the hysteresis phenomenon, where the eradication threshold is lower than the epidemic one [39,42,43]. A similar mechanism could be at play here. However, the identification of hysteresis loops requires a better reconstruction of the attraction basins. While the numerical work presented here provided some preliminary understanding, a deeper mathematical analysis would be needed in this direction. Multistability is, instead, not present in two-pathogen models with a complete mutual exclusion. This dynamical feature emerges, however, in the more general case where strains are allowed to interact upon co-infection [15].

While we did not find stable coexistence among strains in the well-mixed system, coexistence was possible in presence of community structure. In this case, strains can minimize competition for hosts through segregation. Importantly, spatial separation alone is not sufficient to enable coexistence between two strains, when complete mutual exclusion is assumed. This was already known from previous works which showed that community structure must be combined with some level of heterogeneity across communities to enable coexistence, e.g. a strain-specific adaptation to a population or environment to create an ecological niche [35,37,46–48]. Here, communities are homogeneous and coexistence is the result of the interplay between community structure and presence of the cooperative pathogen. When the two strains are seeded in different communities, their interaction occurs after the time lag necessary for one strain to invade the other community. We found that this interval may allow the resident strain to reach the bulk of infections necessary to fend off the invasion. This mechanism is rooted again in the effect of pathogens' frequencies on strain selective advantage. The drivers of strains' coexistence remain an important problem in disease ecology with applications to both vaccination and emergence of anti-microbial resistance. Within-host and population factors have been studied in the past by several modelling investigations. Notably, while coexistence is not possible in models with complete mutual exclusion, this may be enabled in co-infection models [15–18,49]. Other models have addressed environmental and host population features, such as age structure, contact dynamics and spatial organization [19–21,36]. However, little attention has been dedicated to the effect of an additional co-circulating pathogen. Cobey *et al.* studied the interaction between *Haemophilus influenzae* and *Str. pneumoniae* co-circulating strains [50]. Despite the numerous differences between our model and theirs, their work provides results consistent with ours. Namely, the multi-strain dynamics can be affected by another pathogen.

We simulated the three-player dynamics on networks and we obtained phase diagrams that are similar to the continuous-deterministic counterparts. The discrete/stochastic framework, however, allows for observation of the nature of the phase transitions. Several works recently studied the nature of the epidemic transition for two cooperating pathogens, highlighting differences with the single-pathogen case. Cooperation was found to cause discontinuous transitions where the probability of an outbreak and prevalence change abruptly around a critical value of the transmission rate [40,43], akin to other complex contagion mechanisms such as the ones found in social contagion [51,52]. This phenomenon, however, is sensitive to the network topology, with continuous, discontinuous and hybrid, i.e. continuous in the outbreak probability and discontinuous in the prevalence, transitions observed according to the topology of the network [39,40,42,43,53–55]. Here we found rich dynamics as the impact of stochastic effects. These effects were less important when the difference between strains' epidemiological traits was small. Conversely, for a higher difference in cooperative factor, different outcomes are equally probable. Results presented here are preliminary and limited to two

network configurations. Future work should investigate additional network topologies, e.g. a power-law degree distribution, and further values of the network parameters. In addition, more sophisticated numerical analysis (e.g. scaling analysis) would be needed to better classify the nature of the phase transitions.

Concurrence of inter-species cooperation and intra-species competition is present in many epidemiological situations. Currently, around 90 distinct *Str. pneumoniae* serotypes are known to co-circulate worldwide, despite indirect competition mediated by host immune response [4]. The emergence of antibiotic-resistant strains and the development of vaccines able to target only a subset of strains has motivated extensive research on the drivers of *Str. pneumoniae* ecology [4,5,20]. Strain circulation is facilitated by respiratory infections, e.g. influenza [56,57] and some bacterial infections [11,12]. Cooperative behaviour has also been observed between HIV and infections such as HPV, tuberculosis and malaria [1,9,10,58,59]. This increases the burden of these pathogens and causes public health concern. At the same time, there is evidence that different strains of tuberculosis [2,60], malaria [3] and HPV [61–63] may compete. In particular, multidrug-resistant strains of tuberculosis (MDR-TB) are widely spread, although the acquisition of resistance seems to be associated with a fitness cost [59,64]. The synergistic interaction with HIV could play a role in this emergence and surveillance data suggest a possible convergence between HIV and MDR-TB epidemics in several countries [59]. Our theoretical work highlights ecological mechanisms potentially relevant to these examples. In this regard, an essential aspect of our model is the trade-off between transmissibility and cooperativity in determining strain advantage. Although differences in transmissibility across strains have been documented, e.g. fitness cost of resistance [65], gathering information on strain-specific cooperative advantage remains difficult. The theoretical results illustrated here show the importance of quantifying this component for better describing pathogen ecosystems.

This study also represents the starting point of more complex models where multiple strains are involved and competition and cooperation are acting simultaneously. Patterns of competitive and cooperative interactions could be at play, for instance, among recently emerged pathogens such as Zika virus [66]. Zika virus has emerged in regions where Dengue and Chikungunya viruses are endemic. Observed patterns of sequential monodominance by one arbovirus at a time at a given location suggest competition between these pathogens [67]. Also, considerable effort is currently devoted to characterizing possible positive interactions between Zika virus and HIV [66]. In some cases, different strains of the same pathogen can interact both competitively and cooperatively, as in the case of Dengue [8,14]. Primary Dengue infections are characterized by mild symptoms and grant short-term cross-protection against other serotypes. As cross-immunity wanes over time, however, secondary Dengue infections not only become possible but are also associated with severe illness and with increased virulence.

The examples above involve diseases with varying natural history and time scales and should be modelled with different compartmental models—SIS, susceptible-infected-recovered (SIR), susceptible-infected, susceptible-infected-recovered-susceptible [13]. We decided here to consider two SIS pathogens and the results cannot be readily extended to other models, because the dynamics of disease unfolding alters the outcome of strain interactions. It is important to note, however, that several dynamical properties of competitive and cooperative interactions, such as dominance vs. coexistence [27] and abrupt transitions [68–70], hold for both SIS and SIR.

The model studied here is based on certain simplifications. All pathogens are assumed to have the same recovery rate; moreover, cooperation acts in both directions and the same factors $c_i$ quantify the enhancement in susceptibility when $A$ infection occurs before $B_i$ infection and vice versa. These assumptions may not hold for many synergistic pathogens, especially when cooperative benefits are based on different biological mechanisms. For instance, while HIV increases susceptibility against *P. falciparum*, the latter increases HIV's viral load, thus increasing HIV's virulence rather than host susceptibility to HIV [10,58]. It is likely that, by relaxing these assumptions, our model could exhibit even more complex phase diagrams. Eventually, other aspects of the disease-specific mechanisms and multi-pathogen interactions could affect the results presented here and should be addressed in future works. These include latent infections, which are characteristic, for instance, of tuberculosis [2], partial mutual exclusion among strains [2,6,15,16] or interaction mechanisms other than the ones introduced here (e.g. affecting the infectious period [24]).

In conclusion, we have provided a theoretical study of a dynamical system where both competition and cooperation are at play. We found that a less transmissible and more cooperative strain may dominate; however, the conditions on the parameters for this to happen are non-trivial (non-monotonic) and the outcome critically depends on initial conditions and stochastic effects. When coupled with population structure, the presence of a cooperative pathogen may create the conditions

for multi-strain coexistence by dynamically breaking the spatial symmetry and creating ecological niches. These results provide novel ecological insights and suggest mechanisms that may potentially affect the dynamics of interacting epidemics that are of public health concern.

Data accessibility. The python code used for the mean-field analyses and the C++ code for the stochastic simulations on networks are publicly available at the following link: https://github.com/francescopinotti92/Competition-Cooperation.

Authors' contributions. F.P., F.G., P.H. and C.P. conceived and designed the study. F.P. carried out the analyses. F.P., F.G., P.H. and C.P. analysed the results. F.P., F.G., P.H. and C.P. contributed to the writing of the manuscript and approved the final version.

Competing interests. We declare we have no competing interests.

Funding. F.P. acknowledges support by Pierre Louis Doctoral School of Public Health. F.G. and P.H. acknowledge support by German Academic Exchange Service (DAAD) within the PPP-PROCOPE framework (grant no. 57389088). C.P. and F.P. acknowledge support by the Partenariats Hubert Curien within the PPP-PROCOPE (grant no. 35473TK). F.G. acknowledges partial support by Deutsche Forschungsgemeinschaft (DFG) under the grant GH 176/1-1 (idonate project: 345463468). P.H. acknowledges further support by DFG in the framework of Collaborative Research Center 910 (grant no. SFB910). C.P. acknowledges support from the municipality of Paris through the program Emergence(s).

Acknowledgements. We thank Andrew Keane for helpful discussions that made the manuscript more accessible.

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
