## [Reviewer comments · Royal Society Open Science]

Review History

RSOS-190305.R0 (Original submission)

Review form: Reviewer 1

Is the manuscript scientifically sound in its present form?

Yes

Are the interpretations and conclusions justified by the results?

Yes

Is the language acceptable?

Yes

Do you have any ethical concerns with this paper?

No

Have you any concerns about statistical analyses in this paper?

No

Recommendation?

Accept with minor revision (please list in comments)

Comments to the Author(s)

The article “Interplay between competitive and cooperative interactions in multi-pathogen systems” is well written and provides an interesting and novel investigation into the competitive and cooperative dynamics between pathogens and their strains. In addition to combining two previously disparate considerations, the authors should be commended for developing a model that is simple enough to be easily interpretable yet complex enough to allow for interesting dynamics.

A particular strength of the investigation is that a number of different modelling scenarios have been covered: deterministic v. stochastic dynamics; homogeneous v. heterogeneous mixing; and unstructured v. structured populations (e.g. networks). This approach allows for a broad overview of the types of solutions that may arise in the scenarios considered. However, by following this approach it is difficult to provide depth in any one particular area. Reading through I often wondered what type of dynamics may have been observed under alternative parameter configurations. For example, what happens when strain A outcompetes both B strains? Or if A is seeded in the same community as B2 and not B1? Whilst a full exploration of the myriad possibilities would certainly overwhelm the main document, I wonder whether some of these scenarios could be considered, or at least discussed in the appendix, or even alluded to as topics for future research. Some of these cases may even be covered in the articles cited – if so, this should be mentioned.

Nevertheless, I found the article an interesting and thoughtful investigation on multi-strain dynamics that represents a significant advance from earlier investigations.

Minor comments:

P2 L28: Suggest inserting “the” between “assess” and “validity”

P4, equation 1: I think you should mention that the infection rates α and β_i have also been rescaled by μ^{-1} .

P6 Figure 2, caption: Should “C” be lowercase “c”?

P6 L43: Do you mean “infectious population curves” instead of “curves of infectious”?

P10 L47: Suggest inserting “the” between “in” and “presence”

P11 L47-48: Suggest replacing “density of infectious” with “infectious density”

P14 L21: Suggest replacing “This” with “These”

Appendix, Fig S2-3: Although the color-coding can be deduced from the main document, it would be helpful if a separate legend was provided to label the different regions of parameter space.

Review form: Reviewer 2

Is the manuscript scientifically sound in its present form?

Yes

Are the interpretations and conclusions justified by the results?

Yes

Is the language acceptable?

Yes

Do you have any ethical concerns with this paper?

No

Have you any concerns about statistical analyses in this paper?

No

Recommendation?

Accept with minor revision (please list in comments)

Comments to the Author(s)

This is my first reading of this manuscript. Overall, I think this is an interesting study, that should eventually be published. But I have a few comments. This is a theoretical study of infections at the scale of a host population (homogeneously mixed, or subdivided in 2 subpopulations, or in a Erdős-Rényi network, or in a modular network), in the case where there are two competing strains which cannot coinfect the same host, and another strain, which cooperates with the two first strains. One of the first two strains has better cooperation, but a lower transmissibility. The authors study the phase diagrams (which strain(s) manage to maintain in the population) as a function of the relevant parameters.

Around line 14 of page 5 : "We assume then that strain B1 is more cooperative" : maybe say why you don't consider the reverse case. I imagine that you focus on the interesting case of a trade-off between cooperativity and transmissibility, if B2 was both more transmissible and more cooperative, it would outcompete strain B1 (though I am not sure whether there is a range of parameters where coexistence is still possible). Maybe it would be interesting to discuss an example of a pathogen with a trait variation that makes it more transmissible and less cooperative.

About figure 2 : is this possible to get some analytical boundary between the different regimes? The δ_c limit between [A&B2] and [B2] for instance? Or the yellow dotted line? If this is possible, that would be really a significant improvement, the model would enable to understand better what are the crucial parameters.

Figure 2c : have you found numerically initial conditions for which the final state is [A&B2] when δ_c is close to 2 and δ_β is of the order of 0.15?

Figure 4 : initial conditions are not indicated in the legend (and there are some indications in the main text, that there are 100 of each type of infected nodes at first, which is likely important, and should be mentioned in the legend for completeness). For figure 5 the legends should also include the initial conditions.

Around line 38 of page 11 "This may happen also when the transmission rate of the cooperative pathogen 1 is very low or even sub-critical" -> I think it should be added that here, because of the cooperativity, the effective transmission rate of the cooperative pathogen will climb above 1.

Last sentence of page 11 continued on page 12 : It's an interesting idea, but the relation with the results of this manuscript is really unclear to me, there is no concrete example from the results of a situation with no eradication but below the epidemic threshold.

Sentence between the bottom of page 12 and the beginning of page 13 "stochastic effects become less important and the outcome of the competition becomes more net when the difference between the epidemiological traits of the two strains is small" : it is implied that this is an outcome of the small difference, which indeed would be surprising. But actually, I suspect that this is because here, when the difference is small, both are far away from the threshold $\beta=1$, whereas when the difference is large, as the value of the largest β is kept fixed, then the value of the smallest β gets closer to 1, and thus closer to the threshold where the strain is not able to sustain its spread, and thus leading to more stochasticity.

In the next sentence, it is unclear to me what is meant by a "smooth transition".

Appendix, between equations (6) and (7) : why not give a_1 too?

Appendix beginning of section 5 on page 6 "by setting ... =0 it is easy to show" -> I don't think it's actually easy. Besides, in all cases, not only when all the strains coexist, at the equilibrium the time derivatives are zero, so it's unclear what properties are specific to this case. More details on the derivation of the results should be given.

Review form: Reviewer 3

Is the manuscript scientifically sound in its present form?

Yes

Are the interpretations and conclusions justified by the results?

Yes

Is the language acceptable?

Yes

Do you have any ethical concerns with this paper?

No

Have you any concerns about statistical analyses in this paper?

No

Recommendation?

Accept with minor revision (please list in comments)

Comments to the Author(s)

The authors tackle a highly challenging question in epidemiology and public health: co-circulating pathogens, each of which may have multiple strains (and thus result in cooperative or competitive dynamics). These dynamics have strong implications for public health policies on

control and containment, as well as being highly mathematically interesting in their own right due to the high levels of dimensionality in (possibly) non-monotonic systems.

While the qualitative conclusions of the work were not particularly surprising to me, I enjoyed considering the many facets of the authors' investigation. Efforts have clearly been made to consider stochasticity, population structures, and networks (as well as contrasting the different results). As such, I find it to be a valuable contribution to the literature, subject to some revisions listed below.

- I would recommend a thorough revision of the manuscript text, as there are a number of problems with grammar and spelling. This is particularly true in the discussion and conclusion. The content is sound, but I spent many minutes poring over the text before understanding the authors' intended meaning. The section would benefit from re-writing, and perhaps shortening of some of the sentences to increase legibility.

- The authors choose to implement a physically sensible model for two pathogens, A and B. The results section contains a plethora of results which have been obtained numerically. However, at times it is difficult to ascertain the conditions for which each result applies. For example, on page 6, the authors discuss the non-monotonicity of the boundary of $[A \& B ₁]$. I assume this is only true for the case when $\tau < 1$, but it was not quite clear from the text, and other readers may be confused. I understand this becomes more challenging as the dimensionality of the problem increases, but it would be quite helpful to those wishing to intuit what is happening.

- In many of the figures, not all of the regions are explained. For example, in Figure 2 various equilibria are indicated with the analytical boundaries. When looking at the figures, I was missing an explanation as to the hard boundary at $\tau _c = 1$.

- Can the authors please indicate, perhaps in the supplement, the expressions for the various analytical boundaries which are presented? This would complement the numerical results and give a fuller explanation of the various processes.

- The authors have demonstrated the existence and coexistence of the various strains and pathogens. However, it is somewhat inaccurate to claim that these conditions have been given when the results appear to be for a certain subset of parameter space. Naturally, many of these questions require solving high-order polynomials which I understand is cumbersome if not intractable. The authors have shown that many different phenomena may occur, and they point to what they believe to be the driver behind these observations. These are valuable contributions, but it is unclear what the necessary and sufficient conditions are for each of the many cases they discuss.

I am not suggesting they do so, rather that the language be altered to reflect the nature of the findings.

- It would be helpful if the authors could elaborate on the stability of the various states, both locally and the dependence on initial conditions. A number of analytical results are in the supplement, but some comments on stability would be helpful when interpreting the possible implications of the authors' findings.

Review form: Reviewer 4

Is the manuscript scientifically sound in its present form?

Yes

Are the interpretations and conclusions justified by the results?

Yes

Is the language acceptable?

Yes

Do you have any ethical concerns with this paper?

No

Have you any concerns about statistical analyses in this paper?

No

Recommendation?

Major revision is needed (please make suggestions in comments)

Comments to the Author(s)

The paper by Pinotti et al. addresses the interesting question of the interplay between competitive and cooperative interactions in multi-pathogen systems. They focus on a 3-player system with one pathogen A and two pathogen strains B1 and B2 and illustrate through a series of mathematical and simulation results conditions for coexistence and exclusion equilibria between these pathogens, where B1 and B2 cooperate with A but compete directly in an extreme exclusion case with each other. First the authors consider a homogeneous mixing population, secondly they consider a structured population with two groups and relative rates of mixing between them, thirdly they consider explicit contact networks and the effects these may have on the ultimate epidemiological dynamics. Rich phenomenology emerges in each sub-model and the importance of transients and initial conditions is highlighted, especially in the stochastic models. Overall, my impression from this paper is that the authors are being over ambitious and by trying to do too many things at once, they are not really focusing on the key results, they are not providing enough mathematical details and are diluting their message. My suggestion is that sometimes "less is more" and this applies to this paper. My comments are listed below in the hope they can be constructively addressed in a revision.

1. The biological motivation for the study does not seem very well-founded. In the introduction the authors cite a lot the HIV-Tuberculosis coinfection case and the Streptococcus pneumoniae-Influenza case. The first case involves a chronic pathogen SI dynamics for HIV, the second case involves a SIR dynamics for influenza, thus it does not really apply to their model, whose primary epidemiological structure is of SIS type. This is very important as the feedbacks change. Secondly when they talk about coexistence between resistant and sensitive strains, there are several hypotheses in the literature for the mechanisms enabling coexistence and the authors should at least cite some of these before outlining their claim that synergistic interactions with third-parties (other pathogens) may also play a role: the key and very valid point in this paper.

2. The authors are considering an $N=3$ system, and I think they should be explicit about this already in the title, because it makes it clear that already going from $N=2$ (mostly studied in the literature) to $N=3$ allows for much more complex dynamics to emerge. And for $N=3$, one can still obtain some analytical results, which becomes very hard with big N , as most studies perform only simulations in those cases. In this respect I would highly suggest to rephrase the title to 'Interplay between competitive and cooperative interactions in a 3-player pathogen system' or something of the sort. This will make it easier to connect this study also to other ecological studies of the Lotka-Volterra type where such multi-species interaction networks are even more deeply studied, and both mathematical and biological analogies can be exploited.

4. In my view, this study is a special case of certain types of cooperative and competitive interactions. In particular, asymmetric cooperation in co-infection is assumed between 'species': A and B1 (c_1) and A and B2 (c_2) but symmetric competition, and in particular an extreme case of competition ($c=0$) 'within species' B1 and B2 (where co-infection is not allowed). Thus, the model is not very general (see coinfection by the same strain models in Alizon et al 2013, Ecology Letters, and Alizon 2013, J. R.Soc.Interface), and the claim that coexistence between competing

strains is not possible (line 12 in page 12) is not a general one, but only arises here with respect to this particular model structure and the particular assumptions. This should be emphasized, because there are other studies that show that coexistence and even bistability can occur between competing strains depending on the relative magnitudes of within-strain vs. between-strain competition or cooperation (see for example Gjini and Madec 2017, *Theoretical Ecology*) and these studies could be cited. If the authors were to include more general interaction coefficients between B1 and B2 and allow for co-infection by B1 and co-infection by B2 with altered rates, coexistence would be indeed possible, even in the absence of any synergies with third-parties (e.g. A in this study).

5. The presentation of results could be greatly improved. In my opinion, the authors should expand substantially on the mathematical results of section 3.1. and 3.2 and either remove or relegate the sections 3.3-3.4 to the Supplements. I think, the key here is the triangular interaction structure in this '3-species' system (both qualitative and quantitative) and the mathematical criteria determining the biological regimes.

By focusing only on the mean-field scenario and structured population, the paper will be much stronger, clearer and easier to understand. The authors should present their results also in terms of the basic reproduction R_0 of each strain when alone, a quantity that right now is not even mentioned. This will make it easier to relate this study to other epidemiological multi-strain studies. Instead of rescaling time by clearance rate μ , having μ explicit means that many analytical conditions will appear in terms of the strain-specific basic reproduction number R_0 , and the conditions such as: $\alpha > 1$, $\beta_1 > 1$... etc. just become $R_0(A) > 1$ etc. I suggest some of the mathematical results in the Supplements to become part of the main text, especially stability criteria, analytical equilibrium prevalences.

Regions of coexistence and bistability should be studied and presented more in detail, and their biological implications analyzed, as is already done in 3.1-3.2, but somewhat in a brushed over fashion. For example, the last sentence of section 3.1 mentions superficially a very interesting and potentially very important result about multi-stability, but does not describe which parameter regimes lead to such behavior and what the biological implications may be.

I think an important point of section 3.2. is that structured contacts in the host population (already just having 2 sub-populations) allows an internal coexistence equilibrium between 3 strains, which was not possible without host population structure. The authors do not comment on frequency-dependent advantage of each strain, but in fact a lot of the phenomena reported here have to do with frequency-dependence in relative strain fitnesses. Another interesting phenomenon the authors do not comment on is in figure 3d, where intermediate mixing between the two host sub-populations, maximizes the region where A and B1 coexist, thus maximizes the 'rescue' of the less transmissible strain (B1) through its cooperative superiority with a third-party (A). These results deserve to be developed more in depth.

As for Sections 3.3-3.4, the contact network structure, in my view, is just adding more complexity but without providing big new insights. So for me these sections are not necessary. They are rather special cases of a special case and no analytical insights are provided. For example how would the results change if a different n_c were used in 3.4? How would the results change if a different average node degree k were used in 3.3? I don't think these sections add qualitatively much to this paper. Maybe they could be the focus of another paper, focusing specifically on the network structure and studying its effect more in detail. Like this, I feel these sections dilute attention away from the center. How a particular contact network topology modifies this particular assumed interaction structure between 3 strains in my view constitutes another paper. There is much left to explore analytically in the stochasticity, in the discrete nature of events, in the features of the network, and all the rich asymptotic regimes that become possible when

considering the strains interactions. Just illustration of one scenario, as done presently, is not enough.

6. In the interest of clarity and reproducibility, I suggest the authors to compile all model parameter values in a Table by which it can be easier to verify analytic conditions and equilibria that emerge. In particular, and of mathematical importance, how do the absolute values of parameters, namely α , c_1+c_2 , and $\beta_1+\beta_2$ affect the relative competitive dynamics between the 3 strains? The figures focus just on the effect of relative parameter differences, but the absolute values are also very important and deserve some attention. For example in an SIS model (Gjini and Madec 2017), it has been shown that just by changing the value of global R_0 , one can shift the net hierarchy between strains, even when keeping the interaction coefficients the same (i.e. by keeping δ_c or δ_β here the same). It is likely that such effects apply also in this model.

Minor comments:

- I do not understand the need for the extra variables X_i . Cannot they be just incorporated in the force of infection for each strain (dependent on the state of the system), and be put explicitly in the system 1? Having two extra differential equations makes the model cumbersome and adds unnecessary redundancy.

- Please be specific that transmission from co-infected hosts is assumed to happen at the same rate as transmission from single infected hosts, i.e. at rate α for strain A and β_1 and β_2 for strains B1 and B2 from classes D1 and D2. Is this correct?

- I suggest to use word descriptions for the x-y labels in the figures, to recall the variables denoting direct competition (δ_β) between strains B1 and B2 and relative cooperation (δ_c) with the third player A. In fact, ultimately the interplay explored in this paper, is that of direct vs. indirect interactions, which when system size increases further, are likely to generate even richer dynamics.

Decision letter (RSOS-190305.R0)

16-Aug-2019

Dear Dr Poletto,

The editors assigned to your paper ("Interplay between competitive and cooperative interactions in multi-pathogen systems") have now received comments from reviewers. We would like you to revise your paper in accordance with the referee and Associate Editor suggestions which can be found below (not including confidential reports to the Editor). Please note this decision does not guarantee eventual acceptance.

Please submit a copy of your revised paper before 08-Sep-2019. Please note that the revision deadline will expire at 00.00am on this date. If we do not hear from you within this time then it will be assumed that the paper has been withdrawn. In exceptional circumstances, extensions may be possible if agreed with the Editorial Office in advance. We do not allow multiple rounds of revision so we urge you to make every effort to fully address all of the comments at this stage. If deemed necessary by the Editors, your manuscript will be sent back to one or more of the

original reviewers for assessment. If the original reviewers are not available, we may invite new reviewers.

- Data accessibility

If you wish to submit your supporting data or code to Dryad (<http://datadryad.org/>), or modify your current submission to dryad, please use the following link:
<http://datadryad.org/submit?journalID=RSOS&manu=RSOS-190305>

- Competing interests

- Authors' contributions

- Acknowledgements

- Funding statement

on behalf of Dr Berat Haznedaroglu (Associate Editor) and Kevin Padian (Subject Editor)
openscience@royalsociety.org

Editor comments:

The reviewers are all positive about the scope of the study, and they offer a variety of well-considered comments. These may take some while to address and so if you require more time than the schedule typically allows, please notify the editorial office. Be sure to provide detailed responses to all concerns of the reviewers and I would recommend that you have the revised manuscript read by a native Anglophone familiar with the field (and I regret that English is such an irregular language). We will likely have another round of review with one or two of the referees, if they are willing. Best success in your revision.

Comments to Author:

Reviewers' Comments to Author:

Reviewer: 1

Comments to the Author(s)

The article "Interplay between competitive and cooperative interactions in multi-pathogen systems" is well written and provides an interesting and novel investigation into the competitive and cooperative dynamics between pathogens and their strains. In addition to combining two previously disparate considerations, the authors should be commended for developing a model that is simple enough to be easily interpretable yet complex enough to allow for interesting dynamics.

A particular strength of the investigation is that a number of different modelling scenarios have been covered: deterministic v. stochastic dynamics; homogeneous v. heterogeneous mixing; and

unstructured v. structured populations (e.g. networks). This approach allows for a broad overview of the types of solutions that may arise in the scenarios considered. However, by following this approach it is difficult to provide depth in any one particular area. Reading through I often wondered what type of dynamics may have been observed under alternative parameter configurations. For example, what happens when strain A outcompetes both B strains? Or if A is seeded in the same community as B2 and not B1? Whilst a full exploration of the myriad possibilities would certainly overwhelm the main document, I wonder whether some of these scenarios could be considered, or at least discussed in the appendix, or even alluded to as topics for future research. Some of these cases may even be covered in the articles cited – if so, this should be mentioned.

Nevertheless, I found the article an interesting and thoughtful investigation on multi-strain dynamics that represents a significant advance from earlier investigations.

Minor comments:

P2 L28: Suggest inserting “the” between “assess” and “validity”

P4, equation 1: I think you should mention that the infection rates α and β_i have also been rescaled by μ^{-1} .

P6 Figure 2, caption: Should “C” be lowercase “c”?

P6 L43: Do you mean “infectious population curves” instead of “curves of infectious”?

P10 L47: Suggest inserting “the” between “in” and “presence”

P11 L47-48: Suggest replacing “density of infectious” with “infectious density”

P14 L21: Suggest replacing “This” with “These”

Appendix, Fig S2-3: Although the color-coding can be deduced from the main document, it would be helpful if a separate legend was provided to label the different regions of parameter space.

Reviewer: 2

Comments to the Author(s)

This is my first reading of this manuscript. Overall, I think this is an interesting study, that should eventually be published. But I have a few comments. This is a theoretical study of infections at the scale of a host population (homogeneously mixed, or subdivided in 2 subpopulations, or in a Erdős-Rényi network, or in a modular network), in the case where there are two competing strains which cannot coinfect the same host, and another strain, which cooperates with the two first strains. One of the first two strains has better cooperation, but a lower transmissibility. The authors study the phase diagrams (which strain(s) manage to maintain in the population) as a function of the relevant parameters.

Around line 14 of page 5 : “We assume then that strain B1 is more cooperative” : maybe say why you don't consider the reverse case. I imagine that you focus on the interesting case of a trade-off between cooperativity and transmissibility, if B2 was both more transmissible and more cooperative, it would outcompete strain B1 (though I am not sure whether there is a range of parameters where coexistence is still possible). Maybe it would be interesting to discuss an

example of a pathogen with a trait variation that makes it more transmissible and less cooperative.

About figure 2 : is this possible to get some analytical boundary between the different regimes? The Δ_c limit between [A&B2] and [B2] for instance? Or the yellow dotted line? If this is possible, that would be really a significant improvement, the model would enable to understand better what are the crucial parameters.

Figure 2c : have you found numerically initial conditions for which the final state is [A&B2] when Δ_c is close to 2 and Δ_β is of the order of 0.15?

Figure 4 : initial conditions are not indicated in the legend (and there are some indications in the main text, that there are 100 of each type of infected nodes at first, which is likely important, and should be mentioned in the legend for completeness). For figure 5 the legends should also include the initial conditions.

Around line 38 of page 11 "This may happen also when the transmission rate of the cooperative pathogen 1 is very low or even sub-critical" -> I think it should be added that here, because of the cooperativity, the effective transmission rate of the cooperative pathogen will climb above 1.

Last sentence of page 11 continued on page 12 : It's an interesting idea, but the relation with the results of this manuscript is really unclear to me, there is no concrete example from the results of a situation with no eradication but below the epidemic threshold.

Sentence between the bottom of page 12 and the beginning of page 13 "stochastic effects become less important and the outcome of the competition becomes more net when the difference between the epidemiological traits of the two strains is small" : it is implied that this is an outcome of the small difference, which indeed would be surprising. But actually, I suspect that this is because here, when the difference is small, both are far away from the threshold $\beta=1$, whereas when the difference is large, as the value of the largest β is kept fixed, then the value of the smallest β gets closer to 1, and thus closer to the threshold where the strain is not able to sustain its spread, and thus leading to more stochasticity.

In the next sentence, it is unclear to me what is meant by a "smooth transition".

Appendix, between equations (6) and (7) : why not give a_1 too?

Appendix beginning of section 5 on page 6 "by setting ... =0 it is easy to show" -> I don't think it's actually easy. Besides, in all cases, not only when all the strains coexist, at the equilibrium the time derivatives are zero, so it's unclear what properties are specific to this case. More details on the derivation of the results should be given.

Reviewer: 3

Comments to the Author(s)

The authors tackle a highly challenging question in epidemiology and public health: co-circulating pathogens, each of which may have multiple strains (and thus result in cooperative or competitive dynamics). These dynamics have strong implications for public health policies on control and containment, as well as being highly mathematically interesting in their own right due to the high levels of dimensionality in (possibly) non-monotonic systems.

While the qualitative conclusions of the work were not particularly surprising to me, I enjoyed considering the many facets of the authors' investigation. Efforts have clearly been made to

consider stochasticity, population structures, and networks (as well as contrasting the different results). As such, I find it to be a valuable contribution to the literature, subject to some revisions listed below.

- I would recommend a thorough revision of the manuscript text, as there are a number of problems with grammar and spelling. This is particularly true in the discussion and conclusion. The content is sound, but I spent many minutes poring over the text before understanding the authors' intended meaning. The section would benefit from re-writing, and perhaps shortening of some of the sentences to increase legibility.

- The authors choose to implement a physically sensible model for two pathogens, A and B. The results section contains a plethora of results which have been obtained numerically. However, at times it is difficult to ascertain the conditions for which each result applies. For example, on page 6, the authors discuss the non-monotonicity of the boundary of $[A \& B > 1]$. I assume this is only true for the case when $\beta < 1$, but it was not quite clear from the text, and other readers may be confused. I understand this becomes more challenging as the dimensionality of the problem increases, but it would be quite helpful to those wishing to intuit what is happening.

- In many of the figures, not all of the regions are explained. For example, in Figure 2 various equilibria are indicated with the analytical boundaries. When looking at the figures, I was missing an explanation as to the hard boundary at $\beta < c = 1$.

- Can the authors please indicate, perhaps in the supplement, the expressions for the various analytical boundaries which are presented? This would complement the numerical results and give a fuller explanation of the various processes.

- The authors have demonstrated the existence and coexistence of the various strains and pathogens. However, it is somewhat inaccurate to claim that these conditions have been given when the results appear to be for a certain subset of parameter space. Naturally, many of these questions require solving high-order polynomials which I understand is cumbersome if not intractable. The authors have shown that many different phenomena may occur, and they point to what they believe to be the driver behind these observations. These are valuable contributions, but it is unclear what the necessary and sufficient conditions are for each of the many cases they discuss.

I am not suggesting they do so, rather that the language be altered to reflect the nature of the findings.

- It would be helpful if the authors could elaborate on the stability of the various states, both locally and the dependence on initial conditions. A number of analytical results are in the supplement, but some comments on stability would be helpful when interpreting the possible implications of the authors' findings.

Reviewer: 4

Comments to the Author(s)

The paper by Pinotti et al. addresses the interesting question of the interplay between competitive and cooperative interactions in multi-pathogen systems. They focus on a 3-player system with one pathogen A and two pathogen strains B1 and B2 and illustrate through a series of mathematical and simulation results conditions for coexistence and exclusion equilibria between these pathogens, where B1 and B2 cooperate with A but compete directly in an extreme exclusion case with each other. First the authors consider a homogeneous mixing population, secondly they consider a structured population with two groups and relative rates of mixing between them, thirdly they consider explicit contact networks and the effects these may have on the ultimate epidemiological dynamics. Rich phenomenology emerges in each sub-model and the importance of transients and initial conditions is highlighted, especially in the stochastic models. Overall, my impression from this paper is that the authors are being over ambitious and by trying to do too many things at once, they are not really focusing on the key results, they are not providing enough mathematical details and are diluting their message. My suggestion is that sometimes

"less is more" and this applies to this paper. My comments are listed below in the hope they can be constructively addressed in a revision.

1. The biological motivation for the study does not seem very well-founded. In the introduction the authors cite a lot the HIV-Tuberculosis coinfection case and the Streptococcus pneumoniae-Influenza case. The first case involves a chronic pathogen SI dynamics for HIV, the second case involves a SIR dynamics for influenza, thus it does not really apply to their model, whose primary epidemiological structure is of SIS type. This is very important as the feedbacks change. Secondly when they talk about coexistence between resistant and sensitive strains, there are several hypotheses in the literature for the mechanisms enabling coexistence and the authors should at least cite some of these before outlining their claim that synergistic interactions with third-parties (other pathogens) may also play a role: the key and very valid point in this paper.

2. The authors are considering an $N=3$ system, and I think they should be explicit about this already in the title, because it makes it clear that already going from $N=2$ (mostly studied in the literature) to $N=3$ allows for much more complex dynamics to emerge. And for $N=3$, one can still obtain some analytical results, which becomes very hard with big N , as most studies perform only simulations in those cases. In this respect I would highly suggest to rephrase the title to 'Interplay between competitive and cooperative interactions in a 3-player pathogen system' or something of the sort. This will make it easier to connect this study also to other ecological studies of the Lotka-Volterra type where such multi-species interaction networks are even more deeply studied, and both mathematical and biological analogies can be exploited.

4. In my view, this study is a special case of certain types of cooperative and competitive interactions. In particular, asymmetric cooperation in co-infection is assumed between 'species': A and B1 (c_1) and A and B2 (c_2) but symmetric competition, and in particular an extreme case of competition ($c=0$) 'within species' B1 and B2 (where co-infection is not allowed). Thus, the model is not very general (see coinfection by the same strain models in Alizon et al 2013, Ecology Letters, and Alizon 2013, J. R.Soc.Interface), and the claim that coexistence between competing strains is not possible (line 12 in page 12) is not a general one, but only arises here with respect to this particular model structure and the particular assumptions. This should be emphasized, because there are other studies that show that coexistence and even bistability can occur between competing strains depending on the relative magnitudes of within-strain vs. between-strain competition or cooperation (see for example Gjini and Madec 2017, Theoretical Ecology) and these studies could be cited. If the authors were to include more general interaction coefficients between B1 and B2 and allow for co-infection by B1 and co-infection by B2 with altered rates, coexistence would be indeed possible, even in the absence of any synergies with third-parties (e.g. A in this study).

5. The presentation of results could be greatly improved. In my opinion, the authors should expand substantially on the mathematical results of section 3.1. and 3.2 and either remove or relegate the sections 3.3-3.4 to the Supplements. I think, the key here is the triangular interaction structure in this '3-species' system (both qualitative and quantitative) and the mathematical criteria determining the biological regimes.

By focusing only on the mean-field scenario and structured population, the paper will be much stronger, clearer and easier to understand. The authors should present their results also in terms of the basic reproduction R_0 of each strain when alone, a quantity that right now is not even mentioned. This will make it easier to relate this study to other epidemiological multi-strain studies. Instead of rescaling time by clearance rate μ , having μ explicit means that many analytical conditions will appear in terms of the strain-specific basic reproduction number R_0 , and the conditions such as: $\alpha > 1$, $\beta_1 > 1$... etc. just become $R_0(A) > 1$ etc. I suggest some of the

mathematical results in the Supplements to become part of the main text, especially stability criteria, analytical equilibrium prevalences.

Regions of coexistence and bistability should be studied and presented more in detail, and their biological implications analyzed, as is already done in 3.1-3.2, but somewhat in a brushed over fashion. For example, the last sentence of section 3.1 mentions superficially a very interesting and potentially very important result about multi-stability, but does not describe which parameter regimes lead to such behavior and what the biological implications may be.

I think an important point of section 3.2. is that structured contacts in the host population (already just having 2 sub-populations) allows an internal coexistence equilibrium between 3 strains, which was not possible without host population structure. The authors do not comment on frequency-dependent advantage of each strain, but in fact a lot of the phenomena reported here have to do with frequency-dependence in relative strain fitnesses. Another interesting phenomenon the authors do not comment on is in figure 3d, where intermediate mixing between the two host sub-populations, maximizes the region where A and B1 coexist, thus maximizes the 'rescue' of the less transmissible strain (B1) through its cooperative superiority with a third-party (A). These results deserve to be developed more in depth.

As for Sections 3.3-3.4, the contact network structure, in my view, is just adding more complexity but without providing big new insights. So for me these sections are not necessary. They are rather special cases of a special case and no analytical insights are provided. For example how would the results change if a different n_c were used in 3.4? How would the results change if a different average node degree k were used in 3.3? I don't think these sections add qualitatively much to this paper. Maybe they could be the focus of another paper, focusing specifically on the network structure and studying its effect more in detail. Like this, I feel these sections dilute attention away from the center. How a particular contact network topology modifies this particular assumed interaction structure between 3 strains in my view constitutes another paper. There is much left to explore analytically in the stochasticity, in the discrete nature of events, in the features of the network, and all the rich asymptotic regimes that become possible when considering the strains interactions. Just illustration of one scenario, as done presently, is not enough.

6. In the interest of clarity and reproducibility, I suggest the authors to compile all model parameter values in a Table by which it can be easier to verify analytic conditions and equilibria that emerge. In particular, and of mathematical importance, how do the absolute values of parameters, namely α , c_1+c_2 , and $\beta_1+\beta_2$ affect the relative competitive dynamics between the 3 strains? The figures focus just on the effect of relative parameter differences, but the absolute values are also very important and deserve some attention. For example in an SIS model (Gjini and Madec 2017), it has been shown that just by changing the value of global R_0 , one can shift the net hierarchy between strains, even when keeping the interaction coefficients the same (i.e. by keeping δ_c or δ_β here the same). It is likely that such effects apply also in this model.

Minor comments:

- I do not understand the need for the extra variables X_i . Cannot they be just incorporated in the force of infection for each strain (dependent on the state of the system), and be put explicitly in the system 1? Having two extra differential equations makes the model cumbersome and adds unnecessary redundancy.

- Please be specific that transmission from co-infected hosts is assumed to happen at the same rate

as transmission from single infected hosts, i.e. at rate α for strain A and β_1 and β_2 for strains B1 and B2 from classes D1 and D2. Is this correct?

- I suggest to use word descriptions for the x-y labels in the figures, to recall the variables denoting direct competition (δ_{β}) between strains B1 and B2 and relative cooperation (δ_c) with the third player A. In fact, ultimately the interplay explored in this paper, is that of direct vs. indirect interactions, which when system size increases further, are likely to generate even richer dynamics.

Author's Response to Decision Letter for (RSOS-190305.R0)

See Appendix A.

RSOS-190305.R1 (Revision)

Review form: Reviewer 2

Is the manuscript scientifically sound in its present form?

Yes

Are the interpretations and conclusions justified by the results?

Yes

Is the language acceptable?

Yes

Do you have any ethical concerns with this paper?

No

Have you any concerns about statistical analyses in this paper?

No

Recommendation?

Accept with minor revision (please list in comments)

Comments to the Author(s)

I find that the changes in the 3.1 part, detailing more the analytical results delineating the different regimes, is making the manuscript really better. The authors have answered most of my concerns. I still think however that there are aspects that are really confusing.

I think that in terms of spread of pathogens, people are really used to the basic reproduction number R_0 , and that it would be useful to write around line 52 of page 3 explicitly $R_0 = \beta / \mu$ and that, as μ is taken equal to 1 (equivalent to say that time is in units of recovery time) then $R_0 = \beta$. A big issue to me is that then in figure 6, R_0 is not equal to β . In figure 6, if I understand correctly, the average $R_0 = k \beta / \mu = 4 \times 0.015 / 0.05 = 1.2$, which is indeed > 1

and thus supercritical. But why on figure 6, opposite to what is used before, μ is different from 1? Why also not give the explicit formula when discussing supercriticality? Also, the authors added a sentence to answer one of my comments, "Here, the spreading is super-critical for all pathogens: $\beta_1, \beta_2 > 1$ and c_1, c_2 are sufficiently high to sustain the spread of A.", but this sentence is in the paragraph commenting figure 6, where β_1 and β_2 are not > 1 . And it is not even true that $R_0 > 1$ for all strains in the whole figure. Indeed, for $\delta_{\beta} > 0.0025$, then $\beta_1 < 0.0125$ and thus $R_0 < 1$ (though B1 is excluded by B2 before R_0 of B1 gets smaller than 1).

In figure 6, I find panels c, e, g, hard to read, and in principle, there could be cases in which for a given advantage, some simulations give X2 equal to the value taken by X1 in another simulation, preventing them from being both represented on the same graph. Thus it may be better to represent separately X1 and X2.

In legend of figure 2, "In panel (b) transmissibility for B1 is below one for $\delta_{\beta} > 0.1$." "transmissibility" is a bit vague. What not say explicitly β_1 ? And use basic reproduction number instead of transmissibility?

Also, I really like figure 2, and the fact that the boundaries are actually analytical. But then in the legend of figure 2, the numbers referring to the equations in the text corresponding to each type of boundaries should be given in the legend (they are mentioned in the text, but the legend would be much clearer with direct references).

Figure 5 d/e : I think it would be clearer to have these panels combined in 1 (with one set of curves being dashed for instance) to tell immediately how wide is the regime of coexistence.

Legend of figure 7, "Detailed initial conditions are directly shown on each panel." should be completed with something like "the superscript (i) indicate "in community i""

Review form: Reviewer 4

Is the manuscript scientifically sound in its present form?

Yes

Are the interpretations and conclusions justified by the results?

Yes

Is the language acceptable?

Yes

Do you have any ethical concerns with this paper?

No

Have you any concerns about statistical analyses in this paper?

No

Recommendation?

Accept as is

Comments to the Author(s)

I am satisfied with the author's revision of their manuscript and recommend it for publication.

Decision letter (RSOS-190305.R1)

22-Nov-2019

Dear Dr Poletto,

On behalf of the Editors, I am pleased to inform you that your Manuscript RSOS-190305.R1 entitled "Interplay between competitive and cooperative interactions in a three-player pathogen system" has been accepted for publication in Royal Society Open Science subject to minor revision in accordance with the referee suggestions. Please find the referees' comments at the end of this email.

The reviewers and Subject Editor have recommended publication, but also suggest some minor revisions to your manuscript. Therefore, I invite you to respond to the comments and revise your manuscript.

- Ethics statement

- Data accessibility

<http://datadryad.org/submit?journalID=RSOS&manu=RSOS-190305.R1>

- Competing interests

- Authors' contributions

- Acknowledgements

- Funding statement

Because the schedule for publication is very tight, it is a condition of publication that you submit the revised version of your manuscript before 01-Dec-2019. Please note that the revision deadline will expire at 00.00am on this date. If you do not think you will be able to meet this date please let me know immediately.

Supplementary files will be published alongside the paper on the journal website and posted on

the online figshare repository (<https://figshare.com>). The heading and legend provided for each supplementary file during the submission process will be used to create the figshare page, so please ensure these are accurate and informative so that your files can be found in searches. Files on figshare will be made available approximately one week before the accompanying article so that the supplementary material can be attributed a unique DOI.

Kind regards,
Lianne Parkhouse
Editorial Coordinator
Royal Society Open Science
openscience@royalsociety.org

on behalf of Dr Berat Haznedaroglu (Associate Editor) and Kevin Padian (Subject Editor)
openscience@royalsociety.org

Reviewer comments to Author:

Reviewer: 2
Comments to the Author(s)

I find that the changes in the 3.1 part, detailing more the analytical results delineating the different regimes, is making the manuscript really better. The authors have answered most of my concerns. I still think however that there are aspects that are really confusing.

I think that in terms of spread of pathogens, people are really used to the basic reproduction number R_0 , and that it would be useful to write around line 52 of page 3 explicitly $R_0 = \beta / \mu$ and that, as μ is taken equal to 1 (equivalent to say that time is in units of recovery time) then $R_0 = \beta$. A big issue to me is that then in figure 6, R_0 is not equal to β . In figure 6, if I understand correctly, the average $R_0 = k \beta / \mu = 4 \times 0.015 / 0.05 = 1.2$, which is indeed >1 and thus supercritical. But why on figure 6, opposite to what is used before, μ is different from 1? Why also not give the explicit formula when discussing supercriticality? Also, the authors added a sentence to answer one of my comments, "Here, the spreading is super-critical for all pathogens: $\beta_1, \beta_2 > 1$ and c_1, c_2 are sufficiently high to sustain the spread of A.", but this sentence is in the paragraph commenting figure 6, where β_1 and β_2 are not >1 . And it is not even true that $R_0 > 1$ for all strains in the whole figure. Indeed, for $\Delta \beta > 0.0025$, then $\beta_1 < 0.0125$ and thus $R_0 < 1$ (though B1 is excluded by B2 before R_0 of B1 gets smaller than 1).

In figure 6, I find panels c, e, g, hard to read, and in principle, there could be cases in which for a given advantage, some simulations give X2 equal to the value taken by X1 in another simulation, preventing them from being both represented on the same graph. Thus it may be better to represent separately X1 and X2.

In legend of figure 2, "In panel (b) transmissibility for B1 is below one for $\delta \beta > 0.1$." "transmissibility" is a bit vague. What not say explicitly β_1 ? And use basic reproduction number instead of transmissibility?

Also, I really like figure 2, and the fact that the boundaries are actually analytical. But then in the

legend of figure 2, the numbers referring to the equations in the text corresponding to each type of boundaries should be given in the legend (they are mentioned in the text, but the legend would be much clearer with direct references).

Figure 5 d/e : I think it would be clearer to have these panels combined in 1 (with one set of curves being dashed for instance) to tell immediately how wide is the regime of coexistence.

Legend of figure 7, "Detailed initial conditions are directly shown on each panel." should be completed with something like "the superscript (i) indicate "in community i""

Reviewer: 4

Comments to the Author(s)

I am satisfied with the author's revision of their manuscript and recommend it for publication.

Author's Response to Decision Letter for (RSOS-190305.R1)

See Appendices B & C.

Decision letter (RSOS-190305.R2)

13-Dec-2019

Dear Dr Poletto,

It is a pleasure to accept your manuscript entitled "Interplay between competitive and cooperative interactions in a three-player pathogen system" in its current form for publication in Royal Society Open Science.

Due to rapid publication and an extremely tight schedule, if comments are not received, your paper may experience a delay in publication. Royal Society Open Science operates under a continuous publication model. Your article will be published straight into the next open issue and this will be the final version of the paper. As such, it can be cited immediately by other

researchers. As the issue version of your paper will be the only version to be published I would advise you to check your proofs thoroughly as changes cannot be made once the paper is published.

Kind regards,
Lianne Parkhouse
Editorial Coordinator
Royal Society Open Science
openscience@royalsociety.org

on behalf of Dr Berat Haznedaroglu (Associate Editor) and Kevin Padian (Subject Editor)
openscience@royalsociety.org

Appendix A

RSOS-190305: Interplay between competitive and cooperative interactions in a three-player pathogen system

Reviewer 1

The article “Interplay between competitive and cooperative interactions in multi-pathogen systems” is well written and provides an interesting and novel investigation into the competitive and cooperative dynamics between pathogens and their strains. In addition to combining two previously disparate considerations, the authors should be commended for developing a model that is simple enough to be easily interpretable yet complex enough to allow for interesting dynamics.

A particular strength of the investigation is that a number of different modelling scenarios have been covered: deterministic v. stochastic dynamics; homogeneous v. heterogeneous mixing; and unstructured v. structured populations (e.g. networks). This approach allows for a broad overview of the types of solutions that may arise in the scenarios considered. However, by following this approach it is difficult to provide depth in any one particular area. Reading through I often wondered what type of dynamics may have been observed under alternative parameter configurations. For example, what happens when strain A outcompetes both B strains? Or if A is seeded in the same community as B2 and not B1?

Whilst a full exploration of the myriad possibilities would certainly overwhelm the main document, I wonder whether some of these scenarios could be considered, or at least discussed in the appendix, or even alluded to as topics for future research. Some of these cases may even be covered in the articles cited – if so, this should be mentioned.

Response: We thank the reviewer for praising the novelty of the study as well as the relevance of the comparison across modelling frameworks (deterministic/stochastic, homogeneous/heterogeneous mixing).

In order to elaborate on the characterisation of the dynamics, we included additional analyses, which further explore the effect of absolute transmissibilities and cooperation factors. In particular, we performed the stability analysis for the homogenous mixing case and the numerical integration of the system with two communities with additional values of β_2 , c_1

and alpha. Results were presented as multi-panel figures in the Supplementary Material (see Figs. S1 and S3 for the cases of homogeneous mixing and two communities, respectively). Parameter combinations showing qualitative different behaviors were reported in the main paper (figure 2) to allow for discussing the possible dynamical regimes.

We also varied the initial conditions. Specifically, for the homogenous mixing case we better characterised the bistability region with additional numerical analyses. On the case of two communities we analysed additional seeding combinations (B1 in comm 1 and A&B2 in comm2, A&B1 and A&B2, A&B1&B2 in both communities, A&B1&B2 in just one community, B1&B2 and A, A&B1 and B2). In addition, dynamical trajectories were provided together with stable state solutions. This was done by adding one dedicated figure in the Supplementary Material (figure S4), one in the main manuscript (figure 5) and by restructuring previous figure 3, which is now figure 4. We substantially revised the text to improve the presentation of the results. We believe that the revised version provides a clearer comprehension of the multi-pathogen dynamics.

Nevertheless, I found the article an interesting and thoughtful investigation on multi-strain dynamics that represents a significant advance from earlier investigations.

Minor comments:

P2 L28: Suggest inserting “the” between “assess” and “validity”

P4, equation 1: I think you should mention that the infection rates α and β_i have also been rescaled by μ^{-1} .

P6 Figure 2, caption: Should “C” be lowercase “c”?

P6 L43: Do you mean “infectious population curves” instead of “curves of infectious”?

P10 L47: Suggest inserting “the” between “in” and “presence”

P11 L47-48: Suggest replacing “density of infectious” with “infectious density”

P14 L21: Suggest replacing “This” with “These”

Response: We agree and performed all changes requested by the reviewer.

Reviewer: 2

Comments to the Author(s)

This is my first reading of this manuscript. Overall, I think this is an interesting study, that should eventually be published. But I have a few comments. This is a theoretical study of infections at the scale of a host population (homogeneously mixed, or subdivided in 2 subpopulations, or in a Erdős-Rényi network, or in a modular network), in the case where there are two competing strains which cannot coinfect the same host, and another strain, which cooperates with the two first strains. One of the first two strains has better cooperation, but a lower transmissibility. The authors study the phase diagrams (which strain(s) manage to maintain in the population) as a function of the relevant parameters.

Response: We thank the reviewer for her/his comments and we are glad that she/he found the work interesting. We provide a point-by-point response below.

Around line 14 of page 5 : ``We assume then that strain B1 is more cooperative" : maybe say why you don't consider the reverse case. I imagine that you focus on the interesting case of a trade-off between cooperativity and transmissibility, if B2 was both more transmissible and more cooperative, it would outcompete strain B1 (though I am not sure whether there is a range of parameters where coexistence is still possible). Maybe it would be interesting to discuss an example of a pathogen with a trait variation that makes it more transmissible and less cooperative.

Response: Indeed, we decided to focus on this interesting case because a trade-off between cooperativity and transmissibility leads to a rich and complex dynamics. We clarified this point in the Method section:

“Furthermore, we focused on the more interesting case of trade-off between transmissibility and cooperation to limit the parameter exploration: The less transmissible strain, B1, is more cooperative, $\Delta c = c_1 - c_2 > 0$. If B2 is more cooperative, we expect it to win the competition.”

Regarding the biological example, while difference in transmissibility among strains is documented, little information is available regarding strain-specificity in cooperation. We elaborate on this point in the extended Discussion and Conclusion section:

“In this regard, an essential aspect of our model is the trade-off between transmissibility and cooperativity in determining strain advantage. Although differences in transmissibility across strains have been documented, e.g. fitness cost of resistance [64], gathering information on strain-specific cooperative advantage remains difficult. The theoretical results illustrated here show the importance of quantifying this component for better describing pathogen ecosystems”

About figure 2 : is this possible to get some analytical boundary between the different regimes? The Δ_c limit between [A&B2] and [B2] for instance? Or the yellow dotted line? If this is possible, that would be really a significant improvement, the model would enable to understand better what are the crucial parameters.

Response: Analytical expressions were obtained for many of the boundaries of the stability regions. These expressions are now presented and discussed at the beginning of section 3.1.

Figure 2c : have you found numerically initial conditions for which the final state is [A&B2] when Δ_c is close to 2 and Δ_β is of the order of 0.15?

Response: For Δ_c close to 2 and Δ_β close to 0.15 (all other parameters as in figure 2) we have two possible stable states [B2] and [A&B1]. The description of the attraction basins of the two states becomes complex due to the multi-dimensionality of the space of variables.

We explored this more in depth by the numerical integration of the equations. We plotted the final states for $\Delta_c=2$ and $\Delta_\beta=0.15$, as a function of $B_1(0)$ and $B_2(0)$, and considering different values of $A(0)$. We found that for each value of $B_2(0)$ the [A&B1] state is reached when $B_1(0)$ is above a certain threshold. This threshold becomes smaller when $A(0)$ increases. The same analysis was done for alternative values of Δ_c and Δ_β , with similar qualitative results. In addition, we plotted in the parameter space (Δ_c , Δ_β) the threshold values of $B_1(0)$ (other initial frequencies kept fixed) for which the final state is [A&B1], as well as the threshold value of $A(0)$. We summarised these results in a multi-panel figure that was added in the Supplementary Material (figure S2).

Figure 4 : initial conditions are not indicated in the legend (and there are some indications in the main text, that there are 100 of each type of infected nodes at first, which is likely important, and should be mentioned in the legend for completeness). For figure 5 the legends should also include the initial conditions.

Response: We agree and added the initial conditions in the legends of the figures. These are figures 6 and 7 in the revised version of the manuscript.

Around line 38 of page 11 "This may happen also when the transmission rate of the cooperative pathogen 1 is very low or even sub-critical" -> I think tit should be added that here, because of the cooperativity, the effective transmission rate of the cooperative pathogen will climb above 1.

Response: This is now clearly specified.

Last sentence of page 11 continued on page 12 : It's an interesting idea, but the relation with the results of this manuscript is really unclear to me, there is no concrete example from the results of a situation with no eradication but below the epidemic threshold.

Response: We argue that hysteresis loops may be present in our dynamical system. It is true, however, that their reconstruction would require a deeper understanding of the attraction basins. For clarification, we revised the sentence in question:

“Previous works have analysed multistability in two-pathogen models with cooperation in relation to the hysteresis phenomenon, where the eradication threshold is lower than the epidemic one [42, 38, 41]. A similar mechanism could be at play here. However, the identification of hysteresis loops requires a better reconstruction of the attraction basins. While the numerical work presented here provided some preliminary understanding, a deeper mathematical analysis would be needed in this direction.”

Sentence between the bottom of page 12 and the beginning of page 13 “stochastic effects become less important and the outcome of the competition becomes more net when the difference between the epidemiological traits of the two strains is small” : it is implied that this is an outcome of the small difference, which indeed would be surprising. But actually, I suspect that this is because here, when the difference is small, both are far away from the threshold $\beta=1$, whereas when the difference is large, as the value of the largest β is kept fixed, then the value of the smallest β gets closer to 1, and thus closer to the threshold where the strain is not able to sustain its spread, and thus leading to more stochasticity.

Response: We analysed the dynamics in more depth and found that the enhanced fluctuations are linked to an epidemic transition. By increasing the difference in cooperativity, pathogen A goes from persistence to extinction. This is now explained in the Results section, where the paragraph describing the two transitions was modified as follows:

“The transition between [A&B2] and [A&B1] on the left is sharp. Both the probability of one strain winning over the other and the equilibrium prevalence change abruptly for a critical value of δ_c . Here, the spreading is super-critical for all pathogens: $\beta_1, \beta_2 > 1$ and c_1, c_2 are sufficiently high to sustain the spread of A. The transition is due to the trade-off between B1 and B2 growth rates. Conversely, the probability of ending up in the [B2] state rises slowly, driving the gradual transition from the red to the light blue region on the right. This region appears in correspondence of the bistable region of the continuous/deterministic diagram -- figure 2. Here, A undergoes a transition from persistence to extinction, driven by the drop in c_2 (figure S6). This critical regime is characterised by enhanced stochastic fluctuations.”

The distribution of A prevalence are now shown in figure S6 in the Supplementary Material.

In the next sentence, it is unclear to me what is meant by a “smooth transition”.

Response: We wanted to convey the observation of a gradual change, in contrast to an abrupt change. We replaced “smooth” with “gradual” throughout the text.

Appendix, between equations (6) and (7) : why not give a_1 too?

Response: We agree and added the expression for a_1 together with a_0 and a_2 .

Appendix beginning of section 5 on page 6 “by setting ... =0 it is easy to show” -> I don't think it's actually easy. Besides, in all cases, not only when all the strains coexist, at the equilibrium the time derivatives are zero, so it's unclear what properties are specific to this case. More details on the derivation of the results should be given.

Response: We thank the reviewer for pointing this source of confusion. We considered, indeed, the equilibrium conditions. We corrected the sentence in question and added the equations necessary to obtain the result.

Reviewer: 3

Comments to the Author(s)

The authors tackle a highly challenging question in epidemiology and public health: co-circulating pathogens, each of which may have multiple strains (and thus result in cooperative or competitive dynamics). These dynamics have strong implications for public health policies on control and containment, as well as being highly mathematically interesting in their own right due to the high levels of dimensionality in (possibly) non-monotonic systems.

While the qualitative conclusions of the work were not particularly surprising to me, I enjoyed considering the many facets of the authors' investigation. Efforts have clearly been made to consider stochasticity, population structures, and networks (as well as contrasting the different results). As such, I find it to be a valuable contribution to the literature, subject to some revisions listed below.

Response: We thank the reviewer for praising the relevance of our research question and the mathematical interest of the work. We provide below a point-by-point reply to all comments raised.

- I would recommend a thorough revision of the manuscript text, as there are a number of problems with grammar and spelling. This is particularly true in the discussion and conclusion. The content is sound, but I spent many minutes poring over the text before understanding the authors' intended meaning. The section would benefit from re-writing, and perhaps shortening of some of the sentences to increase legibility.

Response: We thoroughly revised the text and considerably improved the readability. We hope that the reviewer will find the new version more accessible and easier to follow.

- The authors choose to implement a physically sensible model for two pathogens, A and B. The results section contains a plethora of results which have been obtained numerically. However, at times it is difficult to ascertain the conditions for which each result applies. For example, on page 6, the authors discuss the non-monotonicity of the boundary of $[A \& B > 1]$. I assume this is only true for the case when $\beta < 1$, but it was not quite clear from the text, and other readers may be confused. I understand this becomes more challenging as the dimensionality of the problem increases, but it would be quite helpful to those wishing to intuit what is happening.

Response: We thank the reviewer for raising this issue. We found that the boundary is monotonous for all tested parameter combinations. We extensively revised the section 3.1 to better describe the dynamics and better illustrate the different regimes - see in particular the discussion around equation (3)-(5) at the beginning of section 3.1. Conditions on the parameters under which the results were recovered are now clearly stated. We also added in

the Supplementary Material a latin square exploration of the parameters beta2, c1 and alpha. See new figure S1 and S3 for the well-mixed and two communities systems, respectively. They summarise all sets of parameters considered.

- In many of the figures, not all of the regions are explained. For example, in Figure 2 various equilibria are indicated with the analytical boundaries. When looking at the figures, I was missing an explanation as to the hard boundary at $c=1$.

Response: We included the explicit expressions of all the boundaries that were computed analytically. These include the boundary for the stability of the [B2] state occurring at $\delta_c=1$ in figure 2a. This boundary is now clearly indicated in the figure. We also added the boundary of the [A&B1] and [A&B2] states and discuss all stability regions in greater detail.

- Can the authors please indicate, perhaps in the supplement, the expressions for the various analytical boundaries which are presented? This would complement the numerical results and give a fuller explanation of the various processes.

Response: We agree and substantially amended the Supplementary Material. Mathematical results were also summarised in section 3.1 of the main paper. See answer above.

- The authors have demonstrated the existence and coexistence of the various strains and pathogens. However, it is somewhat inaccurate to claim that these conditions have been given when the results appear to be for a certain subset of parameter space. Naturally, many of these questions require solving high-order polynomials which I understand is cumbersome if not intractable. The authors have shown that many different phenomena may occur, and they point to what they believe to be the driver behind these observations. These are valuable contributions, but it is unclear what the necessary and sufficient conditions are for each of the many cases they discuss. I am not suggesting they do so, rather that the language be altered to reflect the nature of the findings.

Response: We have revised the text to stress whether a result was based on analytical considerations or on numerical integration. In the latter case, the parameter values, for which the behavior was recovered, are clearly reported. For example, a sentence at the beginning of section 3.1 now reads:

“Specifically we computed explicit analytical expressions for states' feasibility and stability conditions in several cases. Furthermore, we performed extensive numerical simulations in cases where closed expressions were difficult to obtain.”

Furthermore, we now explicitly acknowledge in the Discussion and Conclusion section that the analysis of the impact of initial condition is numerical and preliminary:

“... While the numerical work presented here provided some preliminary understanding, a deeper mathematical analysis would be needed in this direction”

- It would be helpful if the authors could elaborate on the stability of the various states, both locally and the dependence on initial conditions. A number of analytical results are in the supplement, but some comments on stability would be helpful when interpreting the possible implications of the authors' findings.

Response: We agree. The stability conditions are now summarised in the Results section (see answers above). In addition, we improved the numerical analysis of the effect of initial conditions in the bistable and multistable regions by including additional plots in the main text and in the Supplementary Material. Specifically, we numerically integrated the equations for specific sets of parameters. We plotted the final states as a function of $B1(0)$ and $B2(0)$ and considered different values of $A(0)$. We summarised these results in figure 3 in the main manuscript and in figure S2 in the Supplementary Material.

We expanded the Discussion and Conclusion section (e.g., second and third paragraph), where the results are now better summarised and discussed in light of the previous literature.

Reviewer: 4

Comments to the Author(s)

The paper by Pinotti et al. addresses the interesting question of the interplay between competitive and cooperative interactions in multi-pathogen systems. They focus on a 3-player system with one pathogen A and two pathogen strains B1 and B2 and illustrate through a series of mathematical and simulation results conditions for coexistence and exclusion equilibria between these pathogens, where B1 and B2 cooperate with A but compete directly in an extreme exclusion case with each other. First the authors consider a homogeneous mixing population, secondly they consider a structured population with two groups and relative rates of mixing between them, thirdly they consider explicit contact networks and the effects these may have on the ultimate epidemiological dynamics. Rich phenomenology emerges in each sub-model and the importance of transients and initial conditions is highlighted, especially in the stochastic models. Overall, my impression from this paper is that the authors are being over ambitious and by trying to do too many things at once, they are not really focusing on the key results, they are not providing enough mathematical details and are diluting their message. My suggestion is that sometimes "less is more" and this applies to this paper. My comments are listed below in the hope they can be constructively addressed in a revision.

Response: We thank the reviewer for pointing out the interest of our research question and for the many constructive comments. We provide below a point-by-point reply.

1. The biological motivation for the study does not seem very well-founded. In the introduction the authors cite a lot the HIV-Tuberculosis coinfection case and the Streptococcus pneumoniae-Influenza case. The first case involves a chronic pathogen SI dynamics for HIV, the second case involves a SIR dynamics for influenza, thus it does not really apply to their model, whose primary epidemiological structure is of SIS type. This is very important as the feedbacks change.

Response: We thank the reviewer for flagging out this point of unclarity. We aim at showing that the ecological dynamics we modelled - two strains competing in presence of another cooperating pathogen - can be relevant for many pathogens. Therefore we believe it is appropriate to mention infectious diseases with different time scales and natural histories. To better convey this message, we have expanded the review, diversifying the list of pathogens and epidemiological examples. We amended both the Introduction and the Discussion and Conclusion sections. In addition, to avoid confusion we have added the following sentence in the Discussion and Conclusion:

"The examples above involve diseases with varying natural history and time scales and should be modelled with different compartmental models -- SIR, SI, SIS, SIRS. We decided here to

consider two SIS pathogens and the results cannot be readily extended to other models, since the dynamics of disease unfolding alters the outcome of strain interactions.”

Secondly when they talk about coexistence between resistant and sensitive strains, there are several hypotheses in the literature for the mechanisms enabling coexistence and the authors should at least cite some of these before outlining their claim that synergistic interactions with third-parties (other pathogens) may also play a role: the key and very valid point in this paper.

Response: Following the reviewer’s suggestion, we have included additional citations in the Introduction to provide a more general context (see answer to the point below):

Gjini and Madec. Theoretical Ecology, 10(1) 129–141, 2017.

Sofonea, Alizon, and Michalakis. Proc R Soc Lond [Biol], 370(1675), 2015.

Alizon. Interface Focus 3(6), 2013

Alizon, de Roode, Michalakis. Ecology Letters, 16(4), 2013.

In addition, we elaborate on this point in the Discussion and Conclusion section in greater detail:

“The drivers of strains’ co-existence remain an important problem in disease ecology with applications to both vaccination and emergence of antimicrobial resistance. Within-host and population factors have been studied in the past by several modelling investigations. Notably, while co-existence is not possible in models with complete mutual exclusion, this may be enabled in co-infection models [15, 47, 16, 17, 18]. Other models have addressed environmental and host population features, such as age-structure, contacts dynamics and spatial organisation [19, 20, 21, 35]. However, little attention has been dedicated to the effect of an additional co-circulating pathogen.”

2. The authors are considering an N=3 system, and I think they should be explicit about this already in the title, because it makes it clear that already going from N=2 (mostly studied in the literature) to N=3 allows for much more complex dynamics to emerge. And for N=3, one can still obtain some analytical results, which becomes very hard with big N, as most studies perform only simulations in those cases. In this respect I would highly suggest to rephrase the title to 'Interplay between competitive and cooperative interactions in a 3-player pathogen system' or something of the sort. This will make it easier to connect this study also to other ecological studies of the Lotka-Volterra type where such multi-species interaction networks are even more deeply studied, and both mathematical and biological analogies can be exploited.

Response: We thank the reviewer for her/his suggestion and revised the title:

“Interplay between competitive and cooperative interactions in a three-player pathogen system”

Furthermore, we refer to the term “three-player pathogen system” several times throughout the paper.

4. In my view, this study is a special case of certain types of cooperative and competitive interactions. In particular, asymmetric cooperation in co-infection is assumed between 'species': A and B1 (c_1) and A and B2 (c_2) but symmetric competition, and in particular an extreme case of competition ($c=0$) 'within species' B1 and B2 (where co-infection is not allowed). Thus, the model is not very general (see coinfection by the same strain models in Alizon et al 2013, Ecology Letters, and Alizon 2013, J. R.Soc.Interface), and the claim that coexistence between competing strains is not possible (line 12 in page 12) is not a general one, but only arises here with respect to this particular model structure and the particular assumptions. This should be emphasized, because there are other studies that show that coexistence and even bistability can occur between competing strains depending on the relative magnitudes of within-strain vs. between-strain competition or cooperation (see for example Gjini and Madec 2017, Theoretical Ecology) and these studies could be cited. If the authors were to include more general interaction coefficients between B1 and B2 and allow for co-infection by B1 and co-infection by B2 with altered rates, coexistence would be indeed possible, even in the absence of any synergies with third-parties (e.g. A in this study).

Response: We have revised the Discussion and Conclusion section to clarify this point. For example, the text discussing multistability now reads :

“Multistability is, instead, not present in two-pathogen models with complete mutual exclusion. This dynamical feature emerges, however, in the more general case where strains are allowed to interact upon co-infection [15]”

Moreover, we specified in the discussion of co-existence in the model with communities that we deal with the case of complete mutual exclusion:

“Importantly, spatial separation alone is not sufficient for enabling co-existence between two strains, when complete mutual exclusion is assumed”.

We have added the suggested references and a brief discussion. See response to comment 1 above:

“Notably, while co-existence is not possible in models with complete mutual exclusion, this may be enabled in co-infection models [15, 47, 16, 17, 18].”

5. The presentation of results could be greatly improved. In my opinion, the authors should expand substantially on the mathematical results of section 3.1. and 3.2 and either remove or relegate the sections 3.3-3.4 to the Supplements. I think, the key here is the triangular interaction structure in this '3-species' system (both qualitative and quantitative) and the mathematical criteria determining the biological regimes.

Response: We followed the reviewer's remarks and substantially expanded sections 3.1 and 3.2. On the other hand, we feel encouraged by the positive feedback from the other reviewers that the comparison between the two frameworks, continuous/deterministic vs. discrete/stochastic, is interesting and worth reporting in the main text. Therefore, we decided not to cut previous sections 3.3 and 3.4, but instead condense and merge them in a single section "Spreading on networks".

By focusing only on the mean-field scenario and structured population, the paper will be much stronger, clearer and easier to understand. The authors should present their results also in terms of the basic reproduction R_0 of each strain when alone, a quantity that right now is not even mentioned. This will make it easier to relate this study to other epidemiological multi-strain studies. Instead of rescaling time by clearance rate μ , having μ explicit means that many analytical conditions will appear in terms of the strain-specific basic reproduction number R_0 , and the conditions such as: $\alpha > 1$, $\beta_1 > 1$... etc. just become $R_0(A) > 1$ etc.

Response: We agree with the reviewer: Explicitly mentioning R_0 could make it easier to relate our work to other epidemiological multi-strain works. However, in the spirit of dynamical systems (e.g. non-dimensional equations), we decided to keep the time rescaling by μ . To enhance clarity, we explicitly state in the Method section that α and β in fact represent the basic reproductive ratio:

"To simplify the analytical expressions we rescaled time by the average infectious period μ^{-1} , which leads to non-dimensional equations. Then, the parameters β_i and α becomes the basic reproductive ratio of each respective pathogen/strain. This implies that the threshold condition $\beta_i, \alpha > 1$ has to be satisfied in order for the respective player to be able to individually reach an endemic state."

I suggest some of the mathematical results in the Supplements to become part of the main text, especially stability criteria, analytical equilibrium prevalences.

Response: We have extensively revised the Result section. We have added mathematical details and introduced equations that had been included in the Supplementary Material. In particular, we added the expression of the boundary conditions for the stable states and mathematical arguments explaining the conditions of dominance of one strain over the other. The different regions of the stability diagram are now discussed in details as well.

Regions of coexistence and bistability should be studied and presented more in detail, and their biological implications analyzed, as is already done in 3.1-3.2, but somewhat in a brushed over fashion. For example, the last sentence of section 3.1 mentions superficially a very interesting and potentially very important result about multi-stability, but does not describe which parameter regimes lead to such behavior and what the biological implications may be.

Response: We have restructured sections 3.1 and 3.2 and the displayed figures, moving there results originally shown in the Supplementary Material. In particular, the diagram displaying multistability has been revised and is now discussed in depth. Consistently, we have revised figure 2 of the main paper to include this case. Furthermore, we substantially improved the analysis of the impact of initial conditions. We considered specific points of the parameter space corresponding to the bistable and multi-stable regions. We plotted the final states as a function of $B1(0)$ and $B2(0)$ and considered different values of $A(0)$. These results are presented in figure 3 in the main paper and figure S2 in the Supplementary Material. This numerical analysis provides additional insight into the role of $B1$, $B2$ and A and their effect on the dominance of strains. We also substantially revised the discussion of these results in the Discussion and Conclusion section.

I think an important point of section 3.2. is that structured contacts in the host population (already just having 2 sub-populations) allows an internal coexistence equilibrium between 3 strains, which was not possible without host population structure. The authors do not comment on frequency-dependent advantage of each strain, but in fact a lot of the phenomena reported here have to do with frequency-dependence in relative strain fitnesses. Another interesting phenomenon the authors do not comment on is in figure 3d, where intermediate mixing between the two host sub-populations, maximizes the region where A and $B1$ coexist, thus maximizes the 'rescue' of the less transmissible strain ($B1$) through its cooperative superiority with a third-party (A). These results deserve to be developed more in depth.

Response: We thank the reviewer for this comment. We expanded the analysis and the discussion of the coexistence regime. In particular, we looked at strain trajectories for different values of the coupling across communities (epsilon). This informed us on the mechanisms enabling coexistence and the role of strain frequencies. It becomes clear why dominance of $B1$ is maximised at intermediate values of coupling. In fact, with the decrease of epsilon, the persistence of $B1$ is increasingly favoured, i.e. it occurs for a wider region of the parameter space. Persistence of $B1$ may be associated either to its dominance (intermediate values of epsilon) or to co-existence (low values of epsilon). The effect of epsilon on the dominance of $B1$ is now discussed in greater detail. We revised previous figure 3d and moved it to a separate figure (now figure 5). There, we also present the pathogens' trajectories as well as the $B1$ persistence region for different values of epsilon.

As for Sections 3.3-3.4, the contact network structure, in my view, is just adding more complexity but without providing big new insights. So for me these sections are not

necessary. They are rather special cases of a special case and no analytical insights are provided. For example how would the results change if a different n_c were used in 3.4? How would the results change if a different average node degree k were used in 3.3? I don't think these sections add qualitatively much to this paper. Maybe they could be the focus of another paper, focusing specifically on the network structure and studying its effect more in detail. Like this, I feel these sections dilute attention away from the center. How a particular contact network topology modifies this particular assumed interaction structure between 3 strains in my view constitutes another paper. There is much left to explore analytically in the stochasticity, in the discrete nature of events, in the features of the network, and all the rich asymptotic regimes that become possible when considering the strains interactions. Just illustration of one scenario, as done presently, is not enough.

Response: We agree with the reviewer that the study of the three-player dynamics on networks invites future studies. Our manuscript can be seen as a first and important step in this direction. We decided to keep the networks section to highlight the broad applicability of the three-player model. We extended the Discussion and Conclusion to clarify that future work will help to analyse how different network topologies and parameters alter the dynamics:

“Results presented here are preliminary and limited to two network configurations. Future work should investigate additional network topologies (e.g. a power law degree distribution), and values of the network parameters. In addition, more sophisticated numerical analysis (e.g. scaling analysis) would be needed to better classify the nature of the phase transitions.”

6. In the interest of clarity and reproducibility, I suggest the authors to compile all model parameter values in a Table by which it can be easier to verify analytic conditions and equilibria that emerge. In particular, and of mathematical importance, how do the absolute values of parameters, namely α , c_1+c_2 , and $\beta_1+\beta_2$ affect the relative competitive dynamics between the 3 strains? The figures focus just on the effect of relative parameter differences, but the absolute values are also very important and deserve some attention. For example in an SIS model (Gjini and Madec 2017), it has been shown that just by changing the value of global R_0 , one can shift the net hierarchy between strains, even when keeping the interaction coefficients the same (i.e. by keeping δ_c or δ_β here the same). It is likely that such such effects apply also in this model.

Response: We explored absolute level of transmissibility in a latin square fashion and provided the corresponding stability diagrams in two new multi-panel plots in the Supplementary Material, figures S1 and S3. The figures summarise all scenarios explored throughout the paper (including the ones discussed in the main paper) and the corresponding equilibria. The captions and legends clearly report all parameter values.

The reviewer is right that absolute values of parameters affect the net hierarchy between strains. We found that increased levels of transmissibility and cooperation increase the

parameter region of B1 dominance. We added a sentence in the Discussion and Conclusion to highlight this result:

“While dominance depends on the difference in epidemiological traits, we found that variations in the absolute cooperation and transmissibility levels may change the hierarchy between strain -- analogously to [15] -- with a higher spreading potential of either B_i or A favouring the more cooperative strain.”

Minor comments:

- I do not understand the need for the extra variables X_i. Cannot they be just incorporated in the force of infection for each strain (dependent on the state of the system), and be put explicitly in the system 1? Having two extra differential equations makes the model cumbersome and adds unnecessary redundancy.

Response: The variables X_i provide additional insight into the question of stability. The benefit of their introduction becomes more evident in the revised version of the manuscript and the extended Supplementary Material, where we added mathematical details in the Result section. Indeed, the condition for the state [A&B_i] to be stable is that the other pathogen B_j gets extinct, i.e. the growth rate of X_j is negative at the equilibrium.

- Please be specific that transmission from co-infected hosts is assumed to happen at the same rate as transmission from single infected hosts, i.e. at rate alpha for strain A and beta1 and beta2 for strains B1 and B2 from classes D1 and D2. Is this correct?

Response: The reviewer’s understanding is correct. In our three-player model, cooperation arises from an increased susceptibility, while transmission is not affected by past exposures and co-infection events. We clarified this point in the Method section by adding the sentence:

“We assumed that the cooperative interaction does not affect infectivity, thus doubly infected individuals, i.e. infected with both A and B, transmit both diseases at their respective infection rates.”

- I suggest to use word descriptions for the x-y labels in the figures, to recall the variables denoting direct competition (delta_beta) between strains B1 and B2 and relative cooperation (delta_c) with the third player A. In fact, ultimately the interplay explored in this paper, is that of direct vs. indirect interactions, which when system size increases further, are likely to generate even richer dynamics.

Response: We added “cooperative advantage” and “competitive advantage” in the axes labels of figures 2 to 7.

Institut Pierre Louis d'Épidémiologie et de Santé Publique
Pierre Louis Institute of Epidemiology and Public Health

Inserm
La science pour la santé
From science to health

**MÉDECINE
SORBONNE
UNIVERSITÉ**

Unité mixte de recherche en santé n° 1136 (UMR-S 1136)
Directrice : Dominique Costagliola

Dr. Chiara Poletto
Inserm & Sorbonne Université
Paris, France

Paris, November 29, 2019.

Appendix B

Dear Editor,

Please find enclosed the revised version of the manuscript number RSOS-190305.R1 by Pinotti et al.

We revised the manuscript according to the comments of Reviewer 2. In particular we added detailed explanations in the text and in the figure captions to clarify the definition of the basic reproductive number and the time scale of the network model. We also modified the Figure 6 according to the Reviewer suggestion.

We enclose hereafter a point-by-point response to the Reviewer remarks and the marked-up file.

Yours sincerely,

Francesco Pinotti, Fakhteh Ghanbarnejad, Philipp Hövel, Chiara Poletto

Appendix C

Manuscript RSOS-190305.R1

Interplay between competitive and cooperative interactions in a three-player pathogen system

We are grateful for the additional comments, which have helped to further improve the presentation of our findings. Below, we provide a point-by-point response. For the Reviewer's convenience, we first repeat his/her comments and then add a short response.

I find that the changes in the 3.1 part, detailing more the analytical results delineating the different regimes, is making the manuscript really better. The authors have answered most of my concerns. I still think however that there are aspects that are really confusing.

We thank the reviewer for finding the manuscript improved. We provide here below response to all comments.

Comment: I think that in terms of spread of pathogens, people are really used to the basic reproduction number R_0 , and that it would be useful to write around line 52 of page 3 explicitly $R_0 = \beta/\mu$ and that, as μ is taken equal to 1 (equivalent to say that time is in units of recovery time) then $R_0 = \beta$. A big issue to me is that then in figure 6, R_0 is not equal to β . In figure 6, if I understand correctly, the average $R_0 = k \beta / \mu = 4 \times 0.015 / 0.05 = 1.2$, which is indeed >1 and thus supercritical. But why on figure 6, opposite to what is used before, μ is different from 1? Why also not give the explicit formula when discussing supercriticality? Also, the authors added a sentence to answer one of my comments, "Here, the spreading is super-critical for all pathogens: $\beta_1, \beta_2 > 1$ and c_1, c_2 are sufficiently high to sustain the spread of A.", but this sentence is in the paragraph commenting figure 6, where β_1 and β_2 are not >1 . And it is not even true that $R_0 > 1$ for all strains in the whole figure. Indeed, for $\Delta_\beta > 0.0025$, then $\beta_1 < 0.0125$ and thus $R_0 < 1$ (though B1 is excluded by B2 before R_0 of B1 gets smaller than 1).

Response: *We have added the explicit expression for R_0 in the methods. When performing simulations an additional time-scale is introduced, i.e. the time step duration. We took this as time unit and rescaled rates in order to have $\mu \ll \Delta t$, thus preventing discrete-time effects. We have added a detailed explanation in the caption of Figure 6.*

Comment: In figure 6, I find panels c, e, g, hard to read, and in principle, there could be cases in which for a given advantage, some simulations give X_2 equal to the value taken by X_1 in another simulation, preventing them from being both represented on the same graph. Thus it may be better to represent separately X_1 and X_2 .

Response: *We agree with the Reviewer that panels c,e,g are unclear. We have modified the Figure showing the prevalence for each strain in separate panels.*

Comment: In legend of figure 2, "In panel (b) transmissibility for B 1 is below one for

$\delta \beta > 0.1$." "transmissibility" is a bit vague. What not say explicitly β_1 ? And use basic reproduction number instead of transmissibility?

Response: *We followed the suggestion of the reviewer to explicitly refer to β_1 .*

Comment: Also, I really like figure 2, and the fact that the boundaries are actually analytical. But then in the legend of figure 2, the numbers referring to the equations in the text corresponding to each type of boundaries should be given in the legend (they are mentioned in the text, but the legend would be much clearer with direct references).

Response: *We have added the respective equation number in the figure legend.*

Comment: Figure 5 d/e : I think it would be clearer to have these panels combined in 1 (with one set of curves being dashed for instance) to tell immediately how wide is the regime of coexistence.

Response: *We thank the reviewer for the comment and considered his/her suggestion. However, we believe that combining the two bundle of curves would be too confusing. We thus decided to keep the Figure as it is*

Comment: Legend of figure 7, "Detailed initial conditions are directly shown on each panel." should be completed with something like "the superscript (i) indicate "in community i"

Response: *Done*

We thank the Reviewer again for his/her efforts and helpful comments, which were a great support to increase the quality of the manuscript.